# MMScan: A Multi-Modal 3D Scene Dataset with Hierarchical Grounded Language Annotations

**Ruiyuan Lyu**[1,2*], **Jingli Lin**[1,3,7*], **Tai Wang**[1*], **Shuai Yang**[1,4*], **Xiaohan Mao**[1,3], **Yilun Chen**[1], **Runsen Xu**[1,5], **Haifeng Huang**[1,4], **Chenming Zhu**[1,6], **Dahua Lin**[1,5,8], **Jiangmiao Pang**[1†]

[1]Shanghai AI Laboratory, [2]Tsinghua University, [3]Shanghai Jiao Tong University,
[4]Zhejiang University, [5]The Chinese University of Hong Kong, [6]The University of Hong Kong,
[7]Zhiyuan College, Shanghai Jiao Tong University, [8]CPII under InnoHK
[*]Equal contribution    [†]Corresponding author

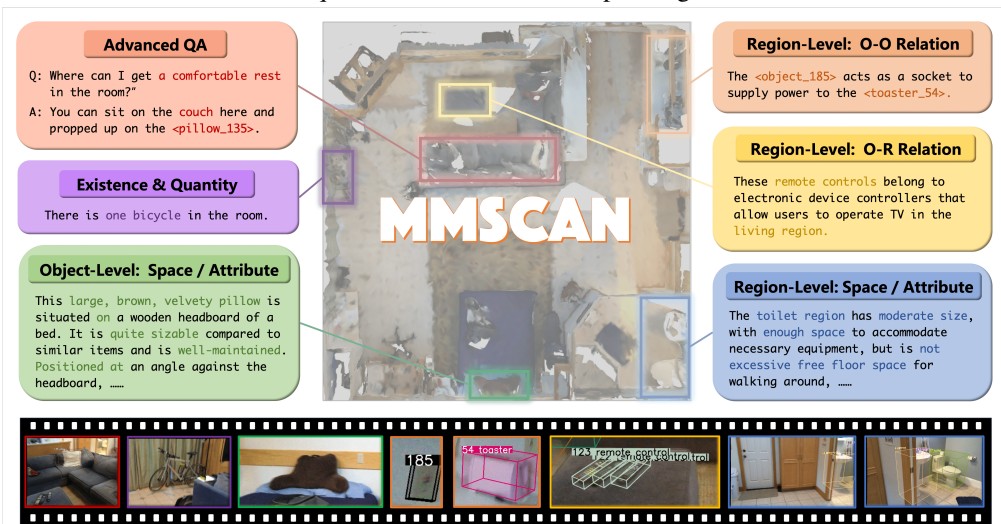

Figure 1: MMScan provides the largest ever multi-modal 3D scene dataset with 6.9M hierarchical grounded language annotations, covering holistic aspects on both object- and region-level.

## Abstract

With the emergence of LLMs and their integration with other data modalities, multi-modal 3D perception attracts more attention due to its connectivity to the physical world and makes rapid progress. However, limited by existing datasets, previous works mainly focus on understanding object properties or inter-object spatial relationships in a 3D scene. To tackle this problem, this paper builds the first largest ever multi-modal 3D scene dataset and benchmark with hierarchical grounded language annotations, MMScan. It is constructed based on a top-down logic, from region to object level, from a single target to inter-target relationships, covering holistic aspects of spatial and attribute understanding. The overall pipeline incorporates powerful VLMs via carefully designed prompts to initialize the annotations efficiently and further involve humans' correction in the loop to ensure the annotations are natural, correct, and comprehensive. Built upon existing 3D scanning data, the resulting multi-modal 3D dataset encompasses 1.4M meta-annotated captions on 109k objects and 7.7k regions as well as over 3.04M diverse samples for 3D visual grounding and question-answering benchmarks. We evaluate representative baselines on our benchmarks, analyze their capabilities in different aspects, and showcase the key problems to be addressed in the future. Furthermore, we use this high-quality dataset to train state-of-the-art 3D visual grounding and LLMs and obtain remarkable performance improvement both on existing benchmarks and in-the-wild evaluation. Codes, datasets, and benchmarks will be available at `https://github.com/OpenRobotLab/EmbodiedScan`.

38th Conference on Neural Information Processing Systems (NeurIPS 2024) Track on Datasets and Benchmarks.

# 1 Introduction

Multi-modal 3D perception is a crucial capability needed by embodied agents and has been extensively studied [12, 6, 28, 15, 50]. As Large Language Models (LLMs) have had great success in recent years, integrating them to build 3D-LLMs is an inevitable trend. However, previous 3D-LLMs [24, 50, 23, 41] can only access object-level datasets [34] or existing limited multi-modal scene datasets [12, 6, 35, 7], thus being constrained to object-level understanding and the recognition of spatial relationships between objects. In contrast, our 3D world has complex hierarchies and rich contexts, suggesting that current 3D-LLMs do not meet our expectations. It urges us to build a comprehensive multi-modal 3D dataset and benchmark to improve the training and evaluation of these models.

To address this problem, this paper aims to build a holistic multi-modal 3D dataset and benchmark at the scene level. Prior works have focused on specific tasks such as 3D visual grounding [12, 6] and question-answering [7, 35] through rule-based or free-form human labeling, resulting in annotations that are either limited to inter-object spatial relationships [6, 47] or influenced by annotators' biases. Recent efforts [57, 29] to expand 3D-language annotations using VLMs have improved scalability but fall short in ensuring correctness and comprehensiveness. Moreover, current annotations lack hierarchical scene structures with fine-grained grounding information, leading to inefficient training for 3D-LLMs and suboptimal instruction following performance.

In contrast to previous works, we introduce a top-down 3D-language annotation logic and present the largest ever multi-modal 3D scene dataset (Fig. 1), *MMScan*, featuring hierarchical language annotations grounded in different granularities of scene context. Constructed using VLM-initialized and human-corrected meta-annotations, the process systematically decomposes complex 3D scenes into region- and object-level instances for comprehensive spatial and attribute annotation. These meta-annotations comprise 1.4M captions over 5k existing real-scanned scenes, 109k objects, and 7.7k regions, forming the basis to produce samples for benchmarks and training.

Given these meta-annotations, we establish two multi-modal 3D perception benchmarks: visual grounding and question-answering. All the samples are generated following two streams, a single target and inter-target relationships and 5 sub-classes (Fig. 3) for different aspects, resulting in 1.28M and 1.76M samples on each benchmark, respectively, to evaluate the model's capabilities from various aspects. In addition, the retained correspondence information of meta-annotations allows us to seamlessly integrate them into scene-level captions. All these caption data, as well as benchmark samples, can serve as a valuable resource for training 3D grounding and large language models.

We evaluate representative baselines on our benchmarks and discuss emerging challenges. Specifically, the performance of visual grounding models is much lower than existing benchmarks, indicating the difficulty of understanding complex prompts that entangle comprehensive spatial and attribute understanding. Including the image modality to enhance semantic understanding and improving the selection of candidate objects are promising directions. For the question-answering benchmark, we observe unsatisfactory results of current 3D-LLMs on our benchmark and a significant improvement, up to 25.6% accuracy, using our data for instruction tuning. Furthermore, we leverage MMScan's captions to train grounding and 3D-LLM models, resulting in a 7.17% AP increase and state-of-the-art performance on existing visual grounding and question-answering benchmarks, more importantly, enabling a much better instruction following performance in the wild.

# 2 Related Work

**Multi-Modal 3D Scene Datasets.** Despite the availability of various 3D scene datasets ranging from the early SUN RGB-D [44] to more recent large-scale ones [21, 9, 46, 11, 42], there remains a scarcity of datasets with multi-modal annotations that focus on language-grounded 3D scene understanding. Predominantly, earlier efforts like ScanRefer [12], ReferIt3D [6], and ScanQA [7] have been centered on ScanNet, pioneering the way of human annotations and template-based generation. SQA3D [35] further emphasizes the role of "situation" in the context. As subsequent efforts, RIORefer [36] and ARKitSceneRefer [32] are similar to ScanRefer but focus on 3RScan and ARKitScene. However, most of them are still limited in amount and scene diversity. Recent initiatives began to pursue scaling up such 3D-text data pairs to push multi-modal 3D learning towards the next stage. For example, 3D-VisTA [57] generates scene descriptions from existing 3D vision-language tasks, templates, and GPT-3 to obtain ScanScribe for pre-training. EmbodiedScan [47] and SceneVerse [29](real& synthetic) either collect more scenes or annotate more objects, scaling up annotations to millions. However, since there are only object annotations in previous scene datasets, all these works lack explicit hierarchical information in 3D scenes, *i.e.*, different granularities of grounding entities.

| Dataset | #Scans | #Language | #Tokens | Correspondence | Focus | Annotation |
|---|---|---|---|---|---|---|
| ScanRefer [12] | 0.7k | 11k | 1.18M | Sent.-Obj. | Natural | Human |
| Nr3D [6] | 0.7k | 42k | 0.62M | Sent.-Obj. | Natural | Human |
| Sr3D [6] | 0.7k | 115k | 1.48M | Sent.-Obj. | OO-Space | Template |
| ScanQA [7] | 0.8k | 41k | - | Sent.-Obj. | QA | AutoGen+Human |
| SQA3D [35] | 0.7k | 53.8k | - | Sent.-Obj. | Situated QA | Human |
| ScanScribe [57] | 1.2k | 278K | 18.49M | Sent.-Obj. | Description | GPT |
| Multi3DRef [52] | 0.7k | 62K | 1.2M | Sent.-Multi-Obj. | Multi-Obj. | GPT+Human |
| EmbodiedScan [47] | 5.2k | 970k | - | Sent.-Obj. | OO-Space | Template |
| RIORefer [36] | 1.4k | 63.6k | 0.94M | Sent.-Obj. | Natural | Human |
| ARKitSceneRefer [32] | 1.6k | 15.6k | 0.22M | Sent.-Obj. | Natural | Human |
| MMScan (Ours) | 5.2k | 6.9M | 114M | Pharse-Obj./Reg. | Holistic | GPT+Temp.+Human |

Table 1: Comparison with other multi-modal 3D real-scanned scene datasets. "Sent.", "Obj.", "Reg.", "OO-Space" and "Temp." refer to "Sentence", "Objects", "Regions", "Object-Object Space" and "Template". MMScan has significant superiority in both the number and quality of annotations.

Furthermore, most works scale up the annotation without humans in the loop, making them not suitable to serve as benchmarks for 3D-LLMs. This paper addresses these gaps and introduces the largest ever multi-modal 3D dataset with comprehensive annotations for both training and benchmarks (Tab. 1).

**Language-Grounded 3D Scene Understanding.** Accompanying these datasets, algorithms for language-grounded 3D scene understanding also make rapid progress. Earlier works focused on crafting specialized models for individual tasks, and recent research has ventured into consolidating these tasks or delving into universal frameworks, capitalizing on the powerful capability of LLMs.

*Specialists for Conventional Tasks.* In the domain of language-grounded 3D scene understanding, traditional tasks encompass: 1) 3D visual grounding [12, 28, 15, 56], which involves identifying 3D objects through 3D bounding boxes or segmentation masks using language cues; 2) 3D question-answering [7, 37, 35], emphasizing the generation of language-based responses; and 3) 3D dense captioning [18, 51, 30, 16], highlighting the synthesis of descriptive language for 3D scenes. Early research was dedicated to refining individual frameworks [28, 53] or modules [15, 56, 45] for these tasks, with some recent efforts exploring task unification [10, 13] and pre-training strategies [57, 31]. Despite these advancements, existing models remain constrained by their task-specific design, hindering broader applicability.

*3D Multi-modal LLMs.* The integration of 3D encoders with LLMs to foster versatile multi-modal 3D intelligence represents an emerging paradigm. Initial efforts [50, 23, 40, 41] have predominantly addressed object-level 3D understanding, leveraging ample 3D-text pairs and simpler configurations. Pioneering work in scene-level comprehension includes 3D-LLM [24], which utilizes pre-trained 2D VLM features, and incorporates positional embeddings and location tokens. Subsequent approaches such as Chat-3D [49] and LL3DA [14] have enabled object-centric interactions through pre-selection techniques. Embodied Generalist [26] underscores the significance of 3D comprehension, delving into the amalgamation of reasoning and action within multi-modal LLMs. Despite the swift evolution of 3D-LLMs, there is an absence of a high-quality, extensive multi-modal 3D scene dataset with hierarchical language annotations. Such a dataset is essential for effectively training these models and for a holistic assessment of their capabilities. This paper endeavors to fill this void, offering a foundational resource for training and evaluating 3D-LLMs.

## 3 Dataset

In this section, we present our approach to building a large-scale 3D scene dataset with hierarchical grounded language annotations. This involves raw data preparation, top-down meta-annotation generation, and extraction of visual grounding and QA task samples. Finally, we make statistics on our annotations and analyze their superiority over existing datasets.

### 3.1 Data Collection & Processing

For constructing a multi-modal 3D dataset, we prioritize selecting a foundational 3D scene dataset with extensive, real-scanned sensor data to minimize the sim-to-real gap and facilitate VLMs' automatic 2D image annotation. In addition, extensive object annotations are important to start the annotations from the object level. We opt for EmbodiedScan, chosen for its comprehensive collection of open-source, real-scanned datasets and its detailed object annotations aided by SAM-assisted labeling tools. Additionally, its Sr3D-based [6] language annotations provide a solid base for spatial relationship descriptions among objects.

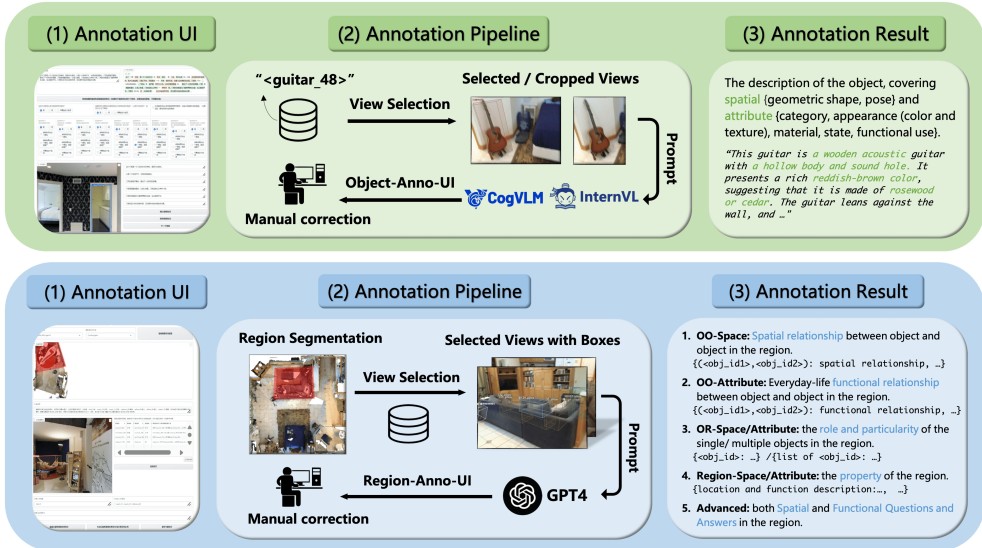

Figure 2: Object-level (top) and region-level (down) meta-annotation UI, pipeline, and examples.

## 3.2 Meta-annotations

We employ a top-down logic to generate comprehensive, hierarchical annotations for scenes, which include non-overlapping captions for scene elements like objects and regions. These *meta-annotations* are designed for broad information coverage and will be further processed to generate specific data samples for benchmarking in Sec. 3.3. We outline our top-down annotation approach and detail the object and region-level language annotations subsequently.

**Overview of Top-Down Logic.** We employ a top-down approach to holistically annotate scenes by segmenting them into regions and objects, potentially including object parts in the future. This method captures information at various levels of granularity. At each level, we solicit human or VLM descriptions of key properties and inter-target relationships, focusing on *spatial* and *attribute* understanding.[1] In this way, we can obtain the scene captions with a hierarchical structure and holistic descriptions for each level. Based on this logic, we will first demonstrate the use of VLMs for object-level language annotation. For the region level, due to the lack of region annotations, we need to annotate regions in each scene and then annotate the key information for each region.

**Object-Level Language Annotation.** We annotate object-level captions based on the bounding boxes from EmbodiedScan. For each object, we establish its main properties, including spatial (geometric shape, pose) and attribute (category, appearance, material, state, functional use) understanding. We then use VLMs with the best image view to initiate descriptions, followed by human annotators refining these with a tailored UI (Fig. 2). Key factors influencing annotation quality are 1) optimal view selection for objects and 2) VLMs selection with appropriate multi-modal prompts for effective caption initialization.

For view selection, we first evaluate image quality by calculating the Laplacian kernel and exclude frames with a variance below 100 to ensure clarity. Next, we project the center and evenly sampled surface points of each object's 3D bounding box onto the image plane. The optimal image is chosen based on the object's center being within the central 25% area and maximizing surface point visibility.

To initialize captions with VLMs, we meticulously craft language prompts and systematically evaluate various visual prompts and VLMs to ensure high-quality descriptions. We find that providing a cropped object patch as a visual prompt

Table 2: VLMs on object-level captioning. "Acc." and "Ann. Ratio" refer to the "accuracy" and "annotation ratio" (some can reject annotation due to security mechanisms). We prioritize "accuracy" here and provide complementary results to achieve a high annotation ratio.

| Models | Free | Acc. | Ann. Ratio |
|---|---|---|---|
| GPT-4v [4] (w/o crop) | ✗ | 84.55% | 87.28% |
| GPT-4v [4] | ✗ | 86.97% | 88.96% |
| Qwen-VL-Max [8] | ✗ | 87.63% | 91.30% |
| InternLM-XComposer [22] | ✓ | 83.63% | 92.86% |
| InternVL-Chat v1.2 [19] | ✓ | 85.75% | 91.06% |
| CogVLM [48] | ✓ | **89.51%** | 85.21% |

---

[1]Since our 3D scene data lacks dynamics, we omit other orthogonal aspects, such as *temporal* and *behavior*.

Table 3: Efficiency comparison of different labeling methods on the object-level annotations. "seconds", "objects", "tokens" are abbreviated as "s", "o", "t".

| Method | Time (s/o) | Avg. Tokens (t/o) |
|--------|-----------|-------------------|
| FM     | 64.5      | 44.6              |
| HIL    | **36.6**  | **85.5**          |

Table 4: Efficiency comparison of different labeling methods on the region-level annotations. "seconds", "sentences", "regions", "tokens" are abbreviated as "s", "sent.", "r", "t".

| Method | Time (s/r) | Avg. Sent. (sent/r) | Avg. Tokens (t/sent) |
|--------|-----------|---------------------|----------------------|
| FM     | 1155.9    | 16.0                | 19.8                 |
| HIL    | **739.0** | **22.0**            | **21.8**             |

yields slightly better results than showing the entire image with 3D bounding boxes. After testing several VLMs on their ability to accurately describe the object properties, we manually checked a subset of 200 objects and determined that CogVLM [48] and InternVL-Chat-V2 [19] perform the best in granularity and accuracy (Tab. 2). Leveraging their complementary strengths, we use the outputs from these two models as initial annotations, which annotators can then modify and refine to enhance the comprehensiveness of the information. See more details on language prompts and additional considerations in the supplementary materials.

**Region Segmentation Annotation.** We've introduced a new UI (Fig. 2) for annotating different regions in each scene, offering a set of predefined categories, including living, study, resting, dining, cooking, bathing, storage, toilet, corridor, open area, others. Annotators are prompted to use 2D polygons to define regions in scenes displayed in a bird's eye view. They can tap on the BEV to access related views of nearby objects, improving annotation accuracy. We eventually amassed 7692 valid annotated regions, excluding "others" and "open area".

**Region-Level Language Annotation.** Based on these region annotations, we further annotate their language descriptions by adapting the object-level language annotation pipeline. Initially, VLMs generate captions from selected views, followed by human revision. For region view selection, we identify target objects and assess their visibility across views based on sharpness and surface points, similar to object view selection. A greedy algorithm is then used to select a minimal set of views to cover all objects. Given the distinct properties of regions, including object-object and object-region relationships, we develop a customized annotation structure and employ varied visual prompts for multi-view images to anchor objects with their identity in captions.

Specifically, we annotate the region for intrinsic property and inter-entity relationships, respectively. Intrinsic properties include location and function, spatial layout and dimensions, architectural elements (doors, windows, walls, floors, ceilings), decorative and soft fitting details, lighting design, color coordination, and style themes, all of which collectively characterize the region. For inter-entity relationships, we annotate 1) object-object relationships, including spatial relationships (based on Sr3D) and attribute relationships, and 2) object-region relationships, such as the role and distinctiveness of the object in the region, to facilitate connections between objects and their respective regions. Finally, we ask annotators to write several advanced QA situated in the scene as a complement.

To anchor objects within captions and ensure consistent object identification across multiple views, we overlay 3D bounding boxes with unique IDs on images as visual cues (Fig. 2). These images are then input into GPT-4 to initiate the annotation process. GPT-4 is instructed to format object references using the template, such as <table 4>, within regional descriptions. This format enables precise object grounding for training and evaluating 3D-LLMs. Note that after preliminary testing with existing VLMs, only GPT-4 can produce reliable captioning with precise object identity information, so we chose it in this round of annotation.

**Efficiency Gains from the GPT-assisted Method.** We have conducted a detailed analysis comparing the annotation efficiency as well as the quality between fully manual annotation and human-in-the-loop annotation. We selected 20 objects and 2 regions from some representative scenes for annotation for this purpose and compared the annotation efficiency quantitatively with similar quality. The comparison includes 1) the annotation time and the number of tokens for each object, measured in seconds/object and tokens/object, 2) the annotation time and the number of sentences for each region, along with the number of tokens per annotation, measured in seconds/region, sentences/region and tokens/sentence. (Tab. 3 and Tab. 4, "fully manual" abbrev. as "FM" , "human-in-the-loop" abbrev. as "HIL"). The comparison results show an overall increase in effective annotations with significantly less time spent using our annotation method.

### 3.3 Post-processing

Given the comprehensive meta-annotations, we subsequently post-process them to obtain data samples for visual grounding and question-answering benchmarks. In addition, we can further

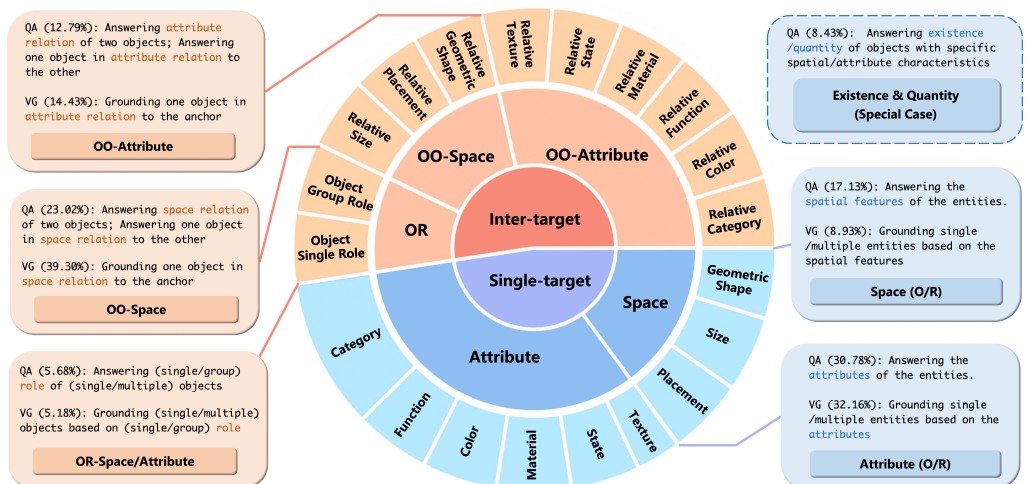

Figure 3: Post-processed annotations for benchmarks. "O" and "R" means "objects" and "regions". Apart from samples shown in the figure, there is a minor part of QA samples for advanced understanding and reasoning, such as situated QA related to everyday life, accounting for 2.18%.

generate grounded scene-level captions from meta-annotations and integrate them to train 3D-LLMs to understand the 3D scene with hierarchical grounding capability more efficiently. Next, we first detail the conversion of meta-annotations to benchmark data samples and then demonstrate the reformatting process for training purposes.

**Post-processing Annotations for Benchmarks.** Meta-annotations have covered most information in each scene, including each object, region, and simple relationships among these entities. Except for simple captioning tasks by directly using meta-annotations, visual grounding and question-answering are two fundamental tasks that necessitate the model to generate responses for more specific targets or questions. As shown in Fig. 3, we categorize the main questions into two cases: asking about a single target or inter-target relationships. For each of them, we similarly target two aspects, space and attribute understanding, on different granularities, objects and regions, to produce data samples for our benchmark. Next, we detail our approach to obtaining samples for different aspects. By default, the following mentioned data samples are converted by ChatGPT [4] and then checked and corrected by humans. Since the main work is information extraction and rephrasing, ChatGPT can complete it well, and humans only need very little effort afterward.

*Single-target.* Given descriptions for each entity, a natural way to convert them into specific questions is by directly asking about the spatial or attribute characteristic of a single target. As shown in the right part of Fig. 3, questions in QA benchmarks expect the model to identify the spatial features or attributes of the entity, while VG benchmarks directly require the model to locate specific entities within a scene. Two cases here are noteworthy. First, there is a kind of question concerning the existence and quantity of a specific target, which cannot be grounded and thus only exists in our QA benchmark. Second, the spatial or attribute characteristics can either be unique for a specific entity in the scene or common among multiple entities. To address this problem efficiently, we employ ChatGPT to extract descriptions for each spatial feature or attribute, *e.g.*, placement: leaning against the wall at a small angle, devise a set of coarse yet orthogonal categories, *e.g.*, placement: {standing upright, piled up, leaning, lying flat, hanging}, and categorize objects into the set to produce samples for understanding common characteristics. Additionally, we generate data samples to understand unique characteristics by combining the original detailed object-level descriptions with human annotations about the particularity of objects in each region.

*Inter-target.* Similarly, inter-target samples can be divided into understanding spatial and attribute relationships among objects and regions. We focus on object-object and object-region relationships for the current dataset due to scene simplicity, excluding region-region pairs. Specifically, the object-object spatial relationships, a well-studied problem, are refined based on EmbodiedScan annotations. Object-object attribute relationships and object-region relationships are derived from preliminary region-level meta-annotations. We utilize templates to create initial samples, refine them, and expand the dataset using ChatGPT. Note that there are two formats for each object-object relationship sub-

class in the QA benchmark (Fig. 3): to directly inquire about the relationship between two objects or to identify a specific object based on its relationship to another object and inquire about its properties.

**Post-processing Annotations for Training.** Beyond post-processing meta-annotations for benchmarks, we also harness these data for training purposes. We present two primary approaches for leveraging this data: (1) generating grounded scene captions to facilitate efficient grounding training and (2) using all the captions for 3D-LLMs instruction tuning.

*Grounded Scene Captions Generation for Efficient Grounding Training.* Previous multi-modal 3D datasets typically generate holistic captions for objects or scenes without hierarchical grounding, lacking detailed correspondence for efficient model training. Drawing on Grounded 3D-LLM [17] and 2D multi-modal learning experiences [38, 43], we retain all correspondence information from meta-annotations. This allows us to seamlessly integrate object- and region-level captions into scene captions, complete with identity information. We log the indices of positive tokens to train the correspondence between 3D features and text phrases. Utilizing this data enhances the model's ability to comprehend complex statements and ground specific objects at the phrase level. Further implementation details are available in the supplementary materials.

*General Captions for Instruction Tuning.* Our meta-annotation offers the most extensive captions for 3D scenes, encompassing various granularities and perspectives for each entity. These captions, along with object- and region-level annotations enriched with grounding information, enable the creation of scene-level captions. They serve as valuable resources for 3D-LLMs' instruction tuning.

Except for these captions used for training, data samples of VG and QA benchmarks are also training resources for instruction tuning. Related attempts are presented in the supplementary materials.

### 3.4 Analysis

**Statistics.** Our dataset comprises 6.9M language annotations and 114M tokens overall, encompassing a comprehensive range of correspondence granularities, as detailed in Tab. 1. It includes 1.4M meta-annotations, with 1.05M specific property captions and 380k complete captions, totaling 18.3M tokens across 109k objects from 285 categories and 7692 regions of 12 types. These meta-annotations facilitate the creation of 1.76M QA samples (4.06M captions), 1.28M VG samples, and 97k grounded scene captions (with 90 tokens per caption), providing various types of data resources for training and benchmarks. See more statistics and distribution figures in the supplemental.

**Comparison to Previous Datasets Annotated with VLMs.** Some recent works like 3D-VisTA[57], LEO[26], Multi3DRef[52], including ours, utilized powerful Visual Language Models for annotation. Among them, MMScan is unique and can bring new insights in the following aspects:

- Top-down annotation logic covering region/object level, single/inter-target descriptions, and spatial/attribute understanding.

- Systematic and customized annotation workflows for objects and regions, including carefully designed language/visual prompts to cover different aspects to obtain meta-annotations instead of direct benchmark samples and detailed ablation for the performance of different Visual Language Models choices.

- Adaptable methods for deriving different benchmark samples from meta-annotations.

- Human-in-the-loop design ensures quality and minimal biases, with UI tools prompting explicit error identification, achieving a sub-5% error rate.

## 4 Experiments

This section presents two main benchmarks based on MMScan: 3D visual grounding and 3D question answering, along with a preliminary 3D captioning benchmark as a potentially more challenging task in the future. We demonstrate new challenges of these tasks with different evaluations from GPT, human, and conventional metrics. Furthermore, we used the rich annotations from MMScan to obtain state-of-the-art grounding models and 3D-LLMs. Finally, we make an analysis of the scaling law of multi-modal 3D learning regarding language annotations. More qualitative and in-the-wild test results and analyses and detailed implementation details can be referred to the appendix.

### 4.1 3D Visual Grounding Benchmark

**Dataset & Evaluation Metrics.** As mentioned previously, we follow the original scene split of EmbodiedScan and obtain 848867/217002/209717 data samples for training/validation/testing on

Table 5: 3D visual grounding benchmark on MMScan.

| Methods | Overall | | | | Single-target | | Inter-target | | |
|---|---|---|---|---|---|---|---|---|---|
| | $AP_{25}$ | $AR_{25}$ | $AP_{50}$ | $AR_{50}$ | ST-attr | ST-space | OO-attr | OO-space | OR |
| ScanRefer [12] | 3.83 | 42.40 | 1.37 | 20.96 | 1.44 | 2.84 | 5.22 | 4.32 | 1.12 |
| BUTD-DETR [28] | 2.29 | 65.61 | 0.84 | 33.11 | 4.79 | 2.04 | 1.49 | 1.75 | 11.87 |
| ViL3DRef [15] | 5.17 | 72.50 | 2.07 | 51.61 | 6.29 | 4.20 | 7.89 | 5.29 | 6.81 |
| ReGround3D [54] | 4.12 | 48.12 | 1.98 | 22.12 | 4.23 | 3.98 | 7.32 | 6.98 | 8.23 |
| MVT [27] | 3.65 | 72.38 | 1.02 | 51.50 | 1.74 | 2.34 | 3.58 | 4.45 | 1.49 |
| 3D-VisTA [57] | 5.24 | **72.51** | 1.91 | **51.85** | 4.91 | 4.39 | 5.75 | 5.99 | 6.35 |
| EmbodiedScan [47] | **10.49** | 47.21 | **2.94** | 21.76 | 7.44 | 7.53 | 13.65 | 11.19 | 7.74 |

our visual grounding benchmark. All the data samples are categorized into a sub-class from the set {ST-attr, ST-space, OO-attr, OO-space, OR}, where *Single-target, attribute, Object-Object, Object-Region* are abbreviated as *ST, attr, OO, OR*. We use 20% samples for training to reduce all the models' training time into 2 days and report the results on the validation set. For the evaluation metric, we adopt the conventional 3D IoU-based Average Precision (AP) used in 3D detection here, considering we can have multiple targets grounded instead of a single one in most previous grounding benchmarks. In addition, we also show the recall and performances for different sub-classes of samples for reference.

**Implementation Details.** We implement four popular baselines, ScanRefer [12], BUTD-DETR [28], ViL3DRef [15], and EmbodiedScan [47], to establish the initial benchmark, considering that they are representative of different types of methods. Note that ViL3DRef requires a prior segmentation of the point cloud as the grounding foundation. We employ the bounding boxes predicted by a trained EmbodiedScan [47] model for point cloud segmentation to ensure a fair comparison. In addition, we replace the RGB-D input of the EmbodiedScan grounding model with the reconstructed 3D point clouds and remove the image feature branch to keep the consistency with other baselines.

**Results.** As shown in Tab. 5, although we train these grounding models with our dataset, the performance is much lower than previous grounding benchmarks (*e.g.*, state-of-the-art 48.1% accuracy on ScanRefer). The new challenges come from diverse and complex prompts that may need to involve LLMs and stronger 3D encoders, more difficult 9-DoF oriented box estimation in EmbodiedScan, and an uncertain number of grounding targets diverging from previous simple settings. In addition, we can find that the single-target performance is typically lower than that of inter-target, indicating that the model can understand inter-target relationships better. We conjecture that there are many small objects and appearance attributes that require involving the image modality to identify. It is also necessary to include image modality as the model design of EmbodiedScan to achieve better performance. We demonstrate several representative failure cases in the supplementary materials. Finally, we observe that ViL3DRef achieves high recall performance due to pretrained detection models on EmbodiedScan but encounters major problems when selecting the correct grounding targets. In contrast, the EmbodiedScan baseline achieves a better selection performance but can further improve the recall capability.

### 4.2 3D Question Answering Benchmark

**Dataset & Evaluation Metrics.** Similarly, following the scene split of EmbodiedScan, our 3D QA benchmark has 1167966/297014/295073 data samples for training/validation/testing correspondingly. Considering the intricacies of questions and answers, and in line with recent practices for assessing LLMs [50, 33, 20], we employ both human evaluators and GPT-4 to ascertain answer accuracy. This approach is complemented by data-driven and conventional metrics. Owing to the prohibitive costs of human evaluation, we primarily present GPT-4's evaluation results in Tab. 6 and demonstrate the concordance between human and GPT-4 assessments on a subset in the supplementary materials.

**Implementation Details.** Given that contemporary 3D-LLMs are anticipated to exhibit generalization in open-world scenarios, we first focus on assessing their zero-shot performance directly on our test set, avoiding the potential overlap of training scenes used by these methods. The baselines consist of 3D-LLM [24], Chat3D-v2 [25], LEO [26], LL3DA [14] and LLaVA-3D [55]. LLaVA-3D is a modified version of PointLLM[55] with RGB-D input to fit scene-level understanding (more details in the supplemental). We utilize the officially released versions of these models, tailor our data and questions to align with their input requirements and evaluate their performance accordingly. Additionally, we fine-tune the models from three recent works, LEO, LL3DA and LLaVA-3D, using our training set to offer reference results under the fine-tuning setting.

Table 6: 3D question answering benchmark on MMScan. "S.-BERT", "B-1", "B-4", "R.-L.", "MET." represents "Sentence-BERT", "BLEU-1", "BLEU-4", "ROUGE-L", "METEOR", respectively. Here, we report the top-1 exact match with (the refined exact-match protocol results) for "EM@1".

| Methods | Setting | Overall | Single-target | | Inter-target | | | Advanced | Data-driven Metrics | | Traditional Metrics | | | | |
|---|---|---|---|---|---|---|---|---|---|---|---|---|---|---|---|
| | | | ST-attr | ST-space | OO-attr | OO-space | OR | | SimCSE | S.-BERT | B-1. | B-4. | R.-L | MET. | EM@1 |
| 3D-LLM [24] | Zero-Shot | 28.6 | 37.8 | 18.8 | 13.7 | 26.3 | 15.4 | 20.8 | 40.4 | 40.3 | 13.4 | 1.5 | 17.3 | 6.0 | 6.2 (19.6) |
| Chat3D-v2 [25] | | 27.9 | 38.1 | 18.3 | 9.3 | 22.4 | 13.5 | 25.4 | 45.4 | 46.3 | 18.0 | 3.0 | 22.9 | 7.5 | 10.2 (19.6) |
| LL3DA [14] | | 15.8 | 15.5 | 14.7 | 14.2 | 25.2 | 4.3 | 6.4 | 40.7 | 43.6 | 5.4 | 2.1 | 16.4 | 4.4 | 8.3 (19.4) |
| LEO [26] | | 22.2 | 28.9 | 17.6 | 18.1 | 20.4 | 15.0 | 16.3 | 40.4 | 41.0 | 11.0 | 0.7 | 17.1 | 4.9 | 9.6 (18.7) |
| LL3DA [14] | Fine-tuning | 38.5 | 40.4 | 46.2 | 14.7 | 47.1 | 26.4 | 7.1 | 65.3 | 67.0 | 26.4 | 8.5 | 44.3 | 14.7 | 30.2 (37.6) |
| LEO [26] | | 47.8 | 55.5 | 49.5 | 36.1 | 45.6 | 32.1 | 38.4 | 71.2 | 72.2 | 32.0 | 12.5 | 52.1 | 17.7 | 36.6 (44.5) |
| LLaVA-3D [55] | | 57.4 | 64.9 | 53.6 | 41.7 | 56.2 | 33.7 | 22.4 | 74.3 | 75.0 | 40.0 | 13.5 | 56.0 | 20.0 | 43.7 (49.5) |

Table 7: Training EmbodiedScan grounding models with MMScan data. "HF" means "Human Fix".

| Methods | HF | Overall | |
|---|---|---|---|
| | | AP$_{25}$ | AP$_{50}$ |
| baseline | - | 37.27 | 17.78 |
| pre-training | ✗ | 42.18 | 21.84 |
| co-training | ✗ | 42.96 | 22.77 |
| pre-training | ✓ | 42.49 | 22.17 |
| co-training | ✓ | 44.44 | 23.69 |

Table 8: Captions tuning of LLaVA-3D on traditional 3D question answering benchmarks.

| Methods | ScanQA (val) | | | | SQA3D (test) |
|---|---|---|---|---|---|
| | B-4. | R.-L. | MET. | EM@1 | EM@1 |
| baseline | 10.5 | 39.2 | 15.1 | 23.1 (39.0) | 51.6 (54.1) |
| + meta-ann. captions | 10.7 | 41.2 | 14.2 | 23.3 (39.3) | 52.7 (54.8) |
| + scene captions | 12.3 | 46.4 | 18.1 | 24.3 (46.6) | 53.2 (55.4) |
| + all captions | 12.7 | 48.1 | 19.8 | 24.7 (48.9) | 54.1 (56.8) |

**Results.** As shown in Tab. 6, the key observation is that fine-tuning with our dataset is necessary and significantly effective, resulting in up to 25.6% accuracy and 27% EM improvement with GPT-4 and EM evaluation (LEO). The results of LL3DA also have a significant improvement. As for zero-shot experiments, we observe much lower performance than expected and a ranking different from previous benchmarks, potentially due to our more comprehensive capability evaluation. For different types of samples, we find the single-target performance is typically higher than that of inter-target, diverging from the visual grounding performance, because previous QA datasets cover more data in this aspect compared to VG datasets. In addition, the single-target performance can be further significantly improved with our data training. For advanced question-answering cases, zero-shot Chat3D-v2 and fine-tuned LLaVA-3D perform much better than others, showing their stronger complex reasoning capabilities.

### 4.3 3D Captioning Benchmark

In addition to the visual grounding and question-answering benchmarks for specific targets, we also build a 3D captioning benchmark based on our meta-annotations. Due to its complexity in evaluation, we present our preliminary attempt here and will further polish the setting in the future.

**Dataset & Evaluation Metrics.** We first split all the objects and regions according to the division of scenes and use the meta-annotations for each object or region as the ground truth. Similarly to the 3D Question Answering benchmark, we employ both human evaluators and GPT-4 to ascertain answer accuracy, complemented by data-driven and conventional metrics, and give the results on the validation set. For the human evaluators, we select 300 samples from the validation set to make the evaluation costs affordable.

**Results.** Similar to fine-tuning with QA samples, we fine-tune LL3DA [14] and LEO [26] and show the results in Tab. 9, the first seven columns in the table represent the scores evaluated by GPT evaluator / humans. We observe that for such complex language tasks, small models such as LL3DA with 1.3B parameters perform much worse than larger ones like LEO.

### 4.4 Analysis

MMScan is a sufficiently challenging dataset benchmark, and beyond employing MMScan for benchmarking state-of-the-art methods, the data can also be leveraged to train stronger models for visual grounding and dialogue tasks. Next, we first demonstrate the challenging nature of the MMScan benchmark, then demonstrate two preliminary attempts to use MMScan data for training and finally present a study regarding the scaling law for multi-modal 3D learning.

**Human Performance Evaluation.** To clearly demonstrate the challenging nature of our MMScan benchmark, we conduct a human performance evaluation . We randomly selected 20 scenes, with one question per sub-category in our benchmark, totaling 120 QA samples to test human performance. We use manual evaluation to ensure the results are accurate. We can see that humans performed much better, and the benchmark is moderately challenging (Tab. 10).

**Grounded Scene Captions for Grounding Training.** As mentioned in Sec. 3.3, we create scene captions with object and region identities to train grounding models effectively. By training the visual

Table 9: 3D captioning benchmark on MMScan.

| Methods | GPT / Human evaluation | | | | | | | Data-driven Metrics | | Traditional Metrics | | | |
|---|---|---|---|---|---|---|---|---|---|---|---|---|---|
| | Overall | Type | Color | Shape | Position | Function | Design | SimCSE | S.-BERT | B-1 | B-4 | MET. | R.-L. |
| LL3DA | 33.6 / 21.1 | 10.0 / 19.2 | 26.3 / 18.0 | 40.6 / 36.0 | 38.9 / 20.2 | 67.5 / 14.6 | 21.7 / 18.1 | 44.9 | 43.5 | 33.6 | 5.3 | 11.9 | 27.2 |
| LEO | 51.3 / 43.2 | 34.9 / 45.2 | 29.7 / 37.0 | 63.0 / 48.7 | 63.7 / 44.1 | 75.0 / 43.8 | 42.7 / 39.2 | 57.1 | 55.2 | 35.5 | 8.3 | 13.8 | 29.5 |

Table 10: Humans' and models' performance on the MMScan QA benchmark (human evaluation).

| Method | Overall | ST-attr | ST-space | OO-attr | OO-space | OR | Advanced |
|---|---|---|---|---|---|---|---|
| LL3DA (finetuned) | 42.8 | 45.0 | 40.0 | 31.6 | 40.0 | 50.0 | 45.0 |
| LEO (finetuned) | 44.6 | 52.6 | 40.0 | 45.0 | 30.0 | 45.0 | 35.0 |
| Human | 77.8 | 85.0 | 68.4 | 66.7 | 77.8 | 84.2 | 70.0 |

grounding baseline on EmbodiedScan with MMScan, we achieve a substantial increase in benchmark performance (up to 7.17% AP), as shown in Tab. 7. Co-training slightly outperforms pre-training, and while human involvement improves data quality, its effect is less pronounced than expected.

**Captions for Instruction Tuning.** A key challenge for current 3D-LLMs to achieve stronger performance is the scarcity of high-quality multi-modal 3D datasets. Utilizing our annotated object and region captions, along with aggregated scene captions, for instruction tuning is a logical approach. We feed these data to our baseline, LLaVA-3D, and observe the significant improvement on traditional question-answering benchmarks (Tab. 8), achieving state-of-the-art performance. Furthermore, it also shows much better in-the-wild test performance, and we present the qualitative results in the supplementary materials.

**Scaling Law for Multi-modal 3D Learning.** Finally, to guide future research, we employ EmbodiedScan VG baseline and LL3DA with different amounts of data to study the scaling law for multi-modal 3D learning. As shown in the upper part of Fig. 4, the VG performance increases steadily while the QA performance exhibits an initial sharp increase followed by a gradual ascent, indicating the VG task still needs more data while our generated QA samples approach saturation. In summary, both tasks show significant improvement with the data increase, from 8.7% to 20.6% AP and 15.84% to 44.81% accuracy on the VG and QA benchmarks, respectively.

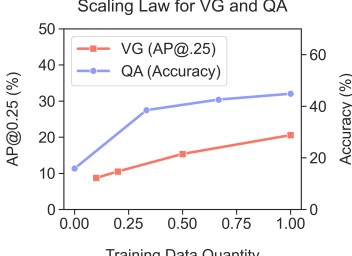

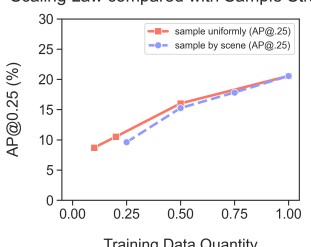

Figure 4: The performance of both tasks grows steadily with the increase of training data, and more diverse scenes can result in more significant improvement.

Furthermore, we conducted an investigation into the scaling law regarding different diversities, *i.e.*, scene diversity vs. data sample diversity. The study evaluated the performance of the models when subjected to distinct sampling methodologies: uniform sampling of data samples and selective sampling based on scenes (with retention of only samples present within the chosen scenes). As depicted in the lower part of Fig. 4, the performance was found to be relatively lower under scenarios with restricted scene diversity, even with an equal distribution of samples. It indicates that both data sample and scene diversity matter when scaling up the training and more diverse scenes can result in more significant improvement.

## 5 Limitations and Conclusion

This paper establishes the largest ever multi-modal 3D scene dataset featuring hierarchical language annotations. We employ a top-down approach and harness both VLMs and human annotators to encompass holistic and precise annotations of 3D scene understanding. Based on meta-annotations, we further derive data samples and grounded scene captions for evaluating and training 3D grounding and language models comprehensively. Although this paper proposes a potentially scalable method to construct large-scale multi-modal 3D datasets, it still relies on human annotators and can be further improved regarding scene diversity. Exploring how to reduce human correction efforts and scale up the scene diversity are objectives for future work.

**Social Impact.** This paper proposes a multi-modal 3D scene dataset based on existing open-source real-scanned data and facilitates the training and evaluation of 3D-LLMs, potentially benefitting downstream 3D content generation and robotic applications. Meanwhile, the trained 3D-LLMs can still have occasional hallucination problems, leading to potential risks when being integrated into the entire system.

**Acknowledgement** This project is funded in part by the National Key R&D Program of China (2022ZD0160201), Shanghai Artificial Intelligence Laboratory, as well as the Centre for Perceptual and Interactive Intelligence (CPII) Ltd under the Innovation and Technology Commission (ITC)'s InnoHK. Dahua Lin is a PI of CPII under the InnoHK.

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

# MMScan: A Multi-Modal 3D Scene Dataset with Hierarchical Grounded Language Annotations

## Appendix

## A   Demo Video

We make a demo video on our project page `https://tai-wang.github.io/mmscan/` for readers to quickly grasp the key idea of this paper. It combs the main content and shows more in-the-wild test results.

## B   Annotation Details

This section supplements several details in the annotation pipeline, including: 1) prompts used for different annotation stages and the implementation of grounded scene captions generation, 2) potential biases in the dataset and how we ensure the validity of the dataset.

### B.1   Meta-annotation

**Object-level Prompts.**   We introduce the subsequent prompt, coupled with a real-captured image, as shown in Fig. 5, to Visual Language Models (VLMs) to initiate the preliminary meta-

annotation process. In accordance with our top-down architectural approach, this meta-annotation is directed to encompass all significant attributes of the object. Within the prompt, the placeholder `{img_object_name}` is intended to be substituted with the actual category label of the object, as determined by ground truth data.

- You are an expert interior designer, who is very sensitive at room furniture and their placements. You are visiting some ordinary rooms that conform to the daily life of an average person. You use your professional expertise to truthfully point out their various aspects. Please describe the single `{img_object_name}` in the center of the image, mainly including the following aspects: appearance (shape, color), material, size (e.g., larger or smaller compared to similar items), condition (e.g., whether a door is open or closed), placement (e.g.,vertical/leaning/slanting/stacked), functionality (compared to similar items), and design features (e.g., whether a chair has armrests/backrest). Please aim for a roughly 200-word description, and avoid using expressions like 'the image' in the description.

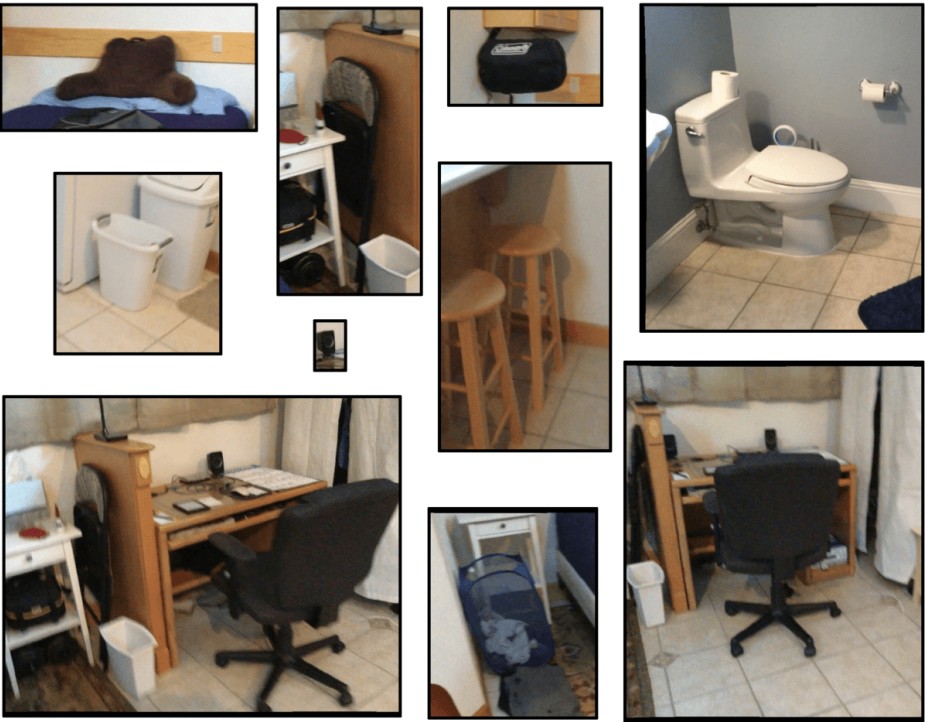

Figure 5: Visual prompts for object-level meta-annotation. The images are cropped to the project area within the object's bounding box after view selection, leading to images of different sizes.

**Region-level Prompts.** To enhance the initial quality of annotation, a structured framework is essential for delineating regions characterized by object-object functional or spatial relationships, as well as for describing the singular or multiple roles of objects within regions, alongside the features of the regions themselves. To facilitate this process, we utilize a multi-stage dialogue with GPT4, requesting structured outputs through API calls with the `json_mode` enabled, which aids in human editing due to the complexity of the factors involved. Additionally, for improved text generation, we incorporate illustrative examples at each point, which are omitted in this prompt. An example of image groups sent to GPT4 can be found in Fig. 6.

- `System Prompt:` You are an expert interior designer, who is very sensitive at room furniture and their placements,and you are particularly familiar with how each piece of furniture relates to everyday human activity. The expected reader is a high-school student with average knowledge of furniture design. I will share photos of a room's <region_type> and use 3D boxes to highlight important items within it. In this region, the items that must be described include <objects_list>, don't leave out any of them.

- I want you to list the location relationships between these objects, where location relationships are between two objects, chosen from <spatial_relation_list>. I want you to provide a JSON file that describes the location relationship of these important items, A dict in the form of '(A,B) : their location relationship'(('<pillow_2>','<bed_1>')):'on'. Here 'on' means that the first item is 'on' the second item).

- Which objects are lying/hanging on the wall? Which objects are standing on the floor? Just pick them out and give me a JSON file.

- I want you to think about are these objects related to each other in the use of functions in everyday activities? Please list the these relationships, in the format of a JSON file. (a dict, the key is a tuple of two items, the value is a sentence describe their relationship)

- What makes the item special in this region? Why did you notice the item? You can think about it in these terms: its special position (like "the chair in the middle of several chairs"), its special role in everyday life (like "I will sit on this chair while eating"). Just give me the result as a JSON file. (a dict, the key is the item, the value is a sentence describe its particularity)

- Which set of these items together belong to a larger class or perform some function together in the region (a set must contain at least two items)? I want you to write this into a JSON file (a dict, the key is a list of items, the value is a sentence. The sentence describes a larger class they belongs to, their role, and their function in the region.

- What's more, based on these layouts, could you share information about the region, these should be included:<Region_features>. Just give me a JSON file. (a dict with keys:<Region_features>,with corresponding values are strings)

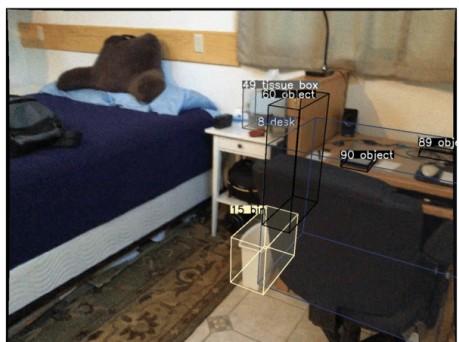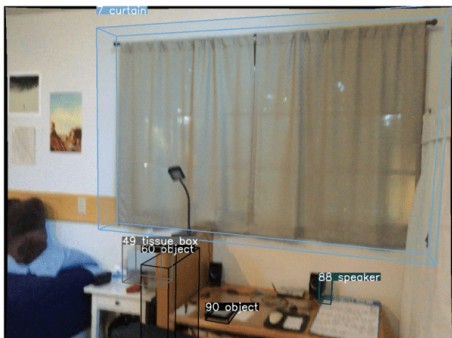
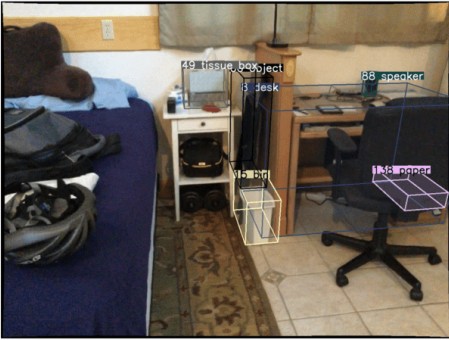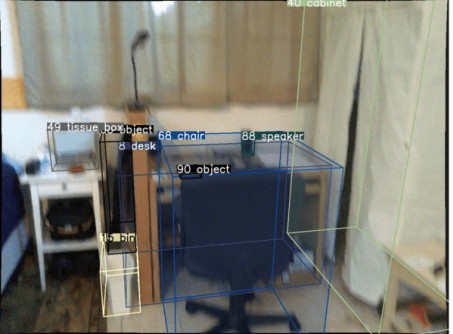

Figure 6: Visual prompts for region-level meta-annotation. Bounding boxes are painted on selected images, accompanied by unique object identifiers and their respective categories. All the images for the region are presented to GPT4 in the first round of conversation.

## B.2 Post-processing

**Prompts for Data Samples Generation.** Both the prompts for 3D Question Answering and 3D Visual Grounding directly or indirectly use the meta-annotations as shown in Fig. 7 and 8, mainly consisting of Single-target Attribute, Single-target Space, (Existence and Quantity), Inter-target Object-Object

**Prompts Generation Examples for 3D QA**

| | |
|---|---|
| **ST Attribute / ST Space** | **Q**: *<bounding box of object X >/<a description without attribute Y of object X >*+ What's the attribute/space P of *<object X>* ?
**A**: *<Unique attribute/space P of object X> / <Common attribute/space P of object X >*

**Q**: *<bounding boxes of objects in region X>*+ What's the attribute/space P of *<region X>* ?
**A**: *<attribute/space P of region X >* |
| **EQ** | **Q**: Is there a *<region/object type>*? / How many *<region/object type>* (with *<attribute/space P >*)?
**A**: *Yes or No / <Number of object/region types>* |
| **IT OO-Attribute** | **Q**: *<bounding box of object X >/<bounding box of object Y >*+ What's the attribute(functional / base attribute P) relationship between *<object X>* and *<object Y>*?
**A**: *<function relationship between ( X , Y )> / <The comparison between (X, Y) in attribute P>*

**Q**: *<bounding box of object X>*+*<object Y>* has *<function relationship between ( X , Y )>* with *<object X>*+What's *<attribute/space P>* of *<object Y>*?
**A**: *<attribute/space P of object Y>* |
| **IT OO-Space** | **Q**: *<bounding box of object X >/<bounding box of object Y >*+ What's the space(spatial position / base space P) relationship between *<object X>* and *<object Y>*?
**A**: *<spatial position relationship between ( X , Y )> / <The comparison between (X, Y) in space P>*

**Q**: *<bounding box of object X>*+*<object Y>* has *<spatial position relationship between ( X , Y )>* with *<object X>*+What's *<attribute/space P>* of *<object Y>*?
**A**: *<attribute/space P of object Y>* |
| **IT OR** | **Q**: *<bounding box of object X>* + What's the *single* role of *<object X> in the region*?
**A**: *<single role description of object X in the region>*

**Q**: *<bounding boxes of objects>* + What's the *multiple* role of *<objects> in the region*?
**A**: *< multiple role description of objects in the region>* |

Figure 7: Prompts for 3D question answering samples generation.

**Prompts Generation Examples for 3D VG**

| | |
|---|---|
| **ST Attribute / ST Space** | 1. *<description about Unique space/attribute of target >* from *<template>* + Find / Choose it.

2. *<Common space/attribute P of target**s**>* + Find / Choose them. |
| **IT OO-Attribute** | 1. (*< short description of anchor >*) + *<The functional relationship between ( anchor, target ) >* + Find / Choose it.

2. (*< short description of anchor >*) + *<The comparison between ( anchor, target ) in attribute P>* + Find / Choose it. |
| **IT OO-Space** | 1. (*< short description of anchor >*) + *<The spatial position relationship between ( anchor, target ) >* + Find / Choose it.

2. (*< short description of anchor >*) + *<The comparison between ( anchor, target ) in space P>* + Find / Choose it. |
| **IT OR** | 1. *<single role description of target in the region>* + Find / Choose it.

2. *<multiple role description of target**s** in the region >* +Find / Choose them. |

Figure 8: Prompts for 3D visual grounding samples generation.

Attribute, Inter-target Object-Object Space, Inter-target Object-Region. Specifically, we use the listed templates to integrate the meta-annotation, obtain *<bounding box>* from the Embodied Scan, *<attribute/space>*, *<description about space/attribute>*, *<Comparison in attribute>* from the Object-level language annotation, and *<functional relationship>*, *<spatial position relationship>*, *<role description>* from the Region-level language annotation.

**Grounded Scene Captions Generation.** The generation of grounded scene captions proposed by Grounded 3D-LLM [17] leverages ChatGPT and 2D vision-language models, utilizing dense object annotations from existing 3D scan datasets. Here, we re-state the details to make the paper self-contained. The process can be divided into 3 main steps:

*Step1. Summarizing object-level meta-annotations.* For each object, we utilize its human-corrected meta-annotations to generate a summarized version with ChatGPT.

*Step2. Condensing objects in local scenes into a caption.* For each enumerated anchor object, we form an initial object set by randomly selecting a group of nearby objects. Their captions and coordinates are fed into GPT-4 for captioning, which is prompted to reference objects by their IDs in the caption.

*Step3. Integrating with rule-based spatial relations.* To enrich scene captions, we integrate program-generated spatial relationships from Sr3D. By selecting an anchor object from the set in Step 2, we apply the spatial relation rules (e.g., between, supporting, nearest, back) to include related objects. GPT-4 then combines these relationships into the prior caption from step 2.

### B.3 Analysis of Potential Biases

Bias in datasets generally stems from uncertain annotation guidelines or situations where subjective judgment is heavily relied upon. To address this, we designed a top-down logic to decompose and refine the annotation requirements, thereby enhancing overall comprehensiveness and reducing freeform annotations. This approach systematically reduces bias compared to previous works. However, some bias is inevitable during annotation. In our screening of VLM and manual annotations, we identified and addressed these biases as follows.

**Bias from VLMs.** Firstly, we will address the bias introduced by the Vision Language Models before the manual annotation and correction.

*Perception Bias.* Bias may occur due to incomplete image inputs (e.g., a cropped image might cause the VLM to focus on the object itself but overlook its relationship with the surrounding environment). To address this, we provide both cropped images and global images with the target object highlighted. At the region level, we supply several images taken from good angles that include all pertinent objects.

*Understanding Bias.* VLMs might carry stereotypes (e.g., frequently describing a chair with "arm-rests" even if the chair does not have any) or misunderstand requirements (e.g., describing unrelated objects). Human annotators correct these by comparing images and revising errors.

*Statement Bias.* VLMs often use uncertain terms like "possibly" or "seems," which are inappropriate for definitive annotations. We screen these with templates and have annotators rewrite vague expressions.

**Bias from Human Annotators.** Secondly, we will address the bias introduced by the manual annotation.

*Perception Bias.* To mitigate the bias similar to that in VLMs, annotators receive multiple images, bird's eye view, and touchpoint display images, as shown in the attached pdf, to help them form a 3D concept of the scene and have enough observations from different perspectives to grasp detailed information.

*Understanding Bias.* Annotators might be careless or make errors (e.g., they might overlook an aspect of an object's appearance, leading to incomplete descriptions, or they might fail to correct errors in VLM results).

*Statement Bias.* Human annotations are expected to be natural and professional, but ambiguities and incoherent expressions can arise when rewriting VLM results (e.g., modifying the description of color in one place but not another, causing ambiguity, or omitting the subject of a sentence after edits). Additionally, since the workers may not be proficient in English, all our user interfaces are designed to support Chinese, helping the workers better understand and express the required annotations, but we must avoid bias during the translation process. We address these issues through multiple rounds of sampling, checking, and feedback.

### B.4 Methods to Ensure the Validity

**Meta-annotations Validity.** To ensure the quality of the final annotations, we adhere to the 95% principle. We randomly inspect 5%-10% of annotations in each screening round. If the pass rate is below 95%, we identify issues, have annotators make corrections, and re-inspect until the standard is met. Finally, we conducted one round of manual annotation and sampling verification for region segmentation. For object-level annotation and region-level annotation, we performed three rounds of manual annotation and sampling verification. Ultimately, each of these met the 95% quality requirement.

**Post-processing Annotations Validity.** For benchmark samples, we derive them following several templates (for different aspects of questions shown in Fig. 3 in the main paper) and then enrich the questions' diversity by refining them with ChatGPT. The overall process introduces very few biases and just relates to reorganizing the information, avoiding obvious risks of yielding factual errors. We verify the logic of derived samples and ensure the overall quality following the previously mentioned random inspection and 95% principle. It turns out that all the samples meet our requirements well.

## C  Implementation Details

This section supplements implementation details for different baselines used in experiments.

### C.1  3D Visual Grounding Baselines

By default, our following re-implemented baselines use the same data augmentation strategy for training. We randomly flip and apply global transformations to the aggregated points and ground-truth boxes, including random rotation with angles in $[-5°, 5°]$, random scaling with a ratio in $[0.9, 1.1]$ and random translation following a normal distribution with standard deviation 0.1.

**ScanRefer [12].** We adapt its officially released code to fit our dataset and experiments. To fit the oriented 3D box output, we add a 6D rotation representation into original regression targets and use a disentangled Chamfer Distance (CD) loss for eight corners to supervise it. We use a pretrained VoteNet [39] detection backbone following the original setting. We use 1 GPU with 32 training samples per batch to train the model for 11 epochs, setting the learning rate to 1e-3 and weight decay to 1e-5.

**BUTD-DETR [28].** Similar to the adaptation of ScanRefer, to fit the oriented 3D box output, we add a three-layer MLP into each prediction head to predict the 6D rotation representation and use a disentangled Chamfer Distance (CD) loss for eight corners to supervise it. We use 8 GPUs with 32 training samples per batch to train the model for 90 epochs, setting the learning rate to 1e-4, the learning rate of the backbone to 1e-3, and weight decay to 1e-5.

**ViL3DRef [15].** ViL3DRef needs instance mask predictions as prior inputs. To fit this requirement, we utilize 3D bounding boxes for point cloud segmentation. Aligning with its original implementation, we use EmbodiedScan's ground truth boxes during training; for testing, we utilize predictions from EmbodiedScan's trained multi-view detection model for a fair comparison. To accommodate the characteristics of multiple targets, we have replaced the single-target cross-entropy loss with its multi-target counterpart. We train the teacher and point encoder models for 25 and 100 epochs, respectively. We use a single GPU with 64 training samples per batch to train the student model for 50 epochs, setting the learning rate to 5e-4, weight decay to 1e-2, and a cosine decay schedule is applied to the learning rate.

**EmbodiedScan [47].** We basically replace the ego-centric views input with point clouds to make it consistent with other baselines. We changed its multi-view image input to point cloud and removed its corresponding ResNet-50 backbone, reducing it to a framework similar to L3Det [56]. Following the original setting, we use a disentangled Chamfer Distance (CD) loss for eight corners to supervise the oriented 3D box output. We use 4 GPUs with 96 training samples per batch to train the model for 12 epochs, setting the learning rate to 5e-4, weight decay to 5e-4, and query number to 100.

**ReGround3D [54].** Since the original ReGround3D supports multi-target grounding, no special architectural modifications to the model are required. We use 8 GPUs with 16 training samples per batch to train the model for 10 epochs. During the training phase, we freeze the vision encoder and apply LoRA to fine-tune the language model (LLM) in the reasoning module. For the 3D grounding module, we freeze the point encoder and train the query selection module and the 3D box decoder. We use the AdamW optimizer with a learning rate of 3e-4, and the training process is guided by a WarmupDecayLR learning rate scheduler with 100 warmup steps.

**MVT [27].** We replaced Cross Entropy Loss in logits loss and language logits loss with Binary Cross Entropy Loss to support multiple ground truths. Following the original setting, we use 1 GPU with 24 training samples per batch to train the model for 24 epochs, setting the base learning rate to 5e-4, the learning rate of the language encoder and refer encoder to 5e-5.

**3D-VisTA [57].** Similarly, we replaced Cross Entropy Loss in logits loss and text logits loss with Binary Cross Entropy Loss to support multiple ground truths. We use 1 GPU with 64 training samples

per batch to train the model for 90 epochs, setting the learning rate to 1e-4, the learning rate of the language encoder to 1e-5.

### C.2   3D Question Answering Baselines

**3D-LLM [24].** We adapt the officially released code and pretrained model. Initially, the point cloud with corresponding features was processed only on Scannet. To fit our dataset, we generate new scans on MP3D and 3RScan. For questions requiring bounding box input, we provided only the question text. Our inference is run using the officially released checkpoint.

**Chat3D-V2 [25].** Instance masks are created using ground truth boxes. Since this pipeline already fits our dataset, no additional adaptation is needed.

**LL3DA [14].** The zero-shot test uses the officially released code and pretrained generalist model. For supervised fine-tuning (SFT), since the model only allows a single object as input, we randomly select one object for questions referring to multiple objects to obtain a visual prompt. The fine-tuning process is performed using two GPUs, with a warming-up rate set to 0.1 epoch.

**LEO [26].**  The zero-shot test uses the officially released code and pretrained generalist model. For SFT, instance masks for object-centric 3D tokens are created using ground truth boxes. When questions refer to multiple objects, we randomly select one to obtain an embodiment token. The fine-tuning process is performed using two GPUs in one epoch, using default settings.

**LLaVA-3D  [55].** LLaVA-3D is an improved version of PointLLM [50]. Due to detailed question-answering and the requirement of understanding the entire scene, we replace the original point cloud encoder with a multi-view image encoder. The multi-view image encoder takes the image features from the pretrained LLaVA and aggregates these patch-wise features by projecting them to 3D space via depth maps. We follow the same downsam- pling strategy in the 3D space to compress these features. We uses LLaVA-3D as the baseline for both the 3D Question Answering benchmark and the Captions for Instruction Tuning experiment.

### C.3   Other Baselines

In the analysis section of the main paper, we conduct experiments to validate the efficacy of training with MMScan. Next, we present the used baselines for two experiments, respectively.

**Grounded Scene Captions for Grounding Training.**  This experiment is mainly based on the EmbodiedScan 3D visual grounding benchmark. We use its officially released baseline without modification, which takes multi-view images as input and grounds target objects given language prompts. Pre-training and co-training experiments both take 12 epochs. The improvement also validates that using more complex multi-modal data for training can also boost the performance of the specific capabilities for inter-object spatial understanding, which is the focus of the original EmbodiedScan's visual grounding data samples.

**Captions for Instruction Tuning.** This experiment is built upon LLaVA-3D, the improved version of PointLLM. Based on this network, we use MMScan for instruction tuning, and it shows significant improvement and achieves state-of-the-art performance on conventional benchmarks, showing the efficacy of training 3D-LLMs with MMScan on previous benchmarks.

## D   Experiments

This section supplements several experiments mentioned in the main paper, including the 3D captioning benchmark based on our meta-annotations, supplementary visual grounding results, using VG and QA samples for instruction tuning, qualitative analysis, and in-the-wild test.

### D.1   Supplementary VG Experiments

**Multi-View Inputs & Detection Pretrained Models.** In accordance with the initial EmbodiedScan approach, we further developed two models utilizing multi-view inputs and subjected them to training. As shown in Tab. 11, training a model directly with multi-view inputs leads to suboptimal results, likely due to the slow convergence of the image backbone. To counteract this issue, we follow the official implementation of EmbodiedScan and use the pretrained detection model to initialize the

Table 11: Analysis on multi-view inputs & using pretrained detection models, where "Recon. Points" means "Reconstructed Points" used in the main paper's benchmark.

| Modality | w/ Pretrain | Overall | | | | Single-target | | Inter-target | | |
| --- | --- | --- | --- | --- | --- | --- | --- | --- | --- | --- |
| | | $AP_{25}$ | $AR_{25}$ | $AP_{50}$ | $AR_{50}$ | ST-attr | ST-space | OO-attr | OO-space | OR |
| Recon. Points | ✗ | 10.49 | 47.21 | 2.94 | 21.76 | 7.44 | 7.53 | 13.65 | 11.19 | 7.74 |
| Multi-View | ✗ | 2.21 | 23.14 | 0.30 | 8.00 | 1.43 | 1.59 | 3.52 | 2.36 | 1.23 |
| Multi-View | ✓ | **14.75** | **48.05** | **4.28** | 21.41 | 11.71 | 11.03 | 16.80 | 15.78 | 9.24 |

Table 12: Captions tuning for 3DLLMs (LEO,LL3DA and the improved PointLLM) on traditional 3D question answering benchmarks. (EM@1 metric)

| Method | ScanQA (val) | SQA3D (test) |
| --- | --- | --- |
| LLaVA-3D (baseline) | 23.1 | 51.6 |
| LLaVA-3D (tuning) | 24.7 | 52.7 |
| LEO (baseline) | 16.94 | 48.50 |
| LEO (tuning) | 17.54 | 51.13 |
| LL3DA (baseline) | 15.61 | 45.11 |
| LL3DA (tuning) | 16.83 | 45.35 |

network. It turns out that such initialization is critical for current visual grounding models and yields even better results than using reconstructed point clouds.

## D.2 Supplementary Caption Tuning Experiments

We have observed significant improvements on traditional question-answering benchmarks after tuning LLaVA-3D with MMScan captions. Here, we demonstrate that tuning with MMScan captions also enhances the performance of LEO and LL3DA on these benchmarks, as shown in the Tab. 12, which confirms the benefits of large-scale MMScan annotation.

## D.3 Using VG and QA for Instruction Tuning

In addition to using caption data for tuning 3D-LLMs, we can also use the VG and QA samples for tuning. Taking the same traditional benchmarks as the example, as shown in Tab. 13, using VG and QA samples for tuning can both bring performance improvement and QA samples are more important than VG and captioning data considering the data format is more consistent with the validation benchmark.

## D.4 Qualitative Analysis

**Qualitative Results.** As shown in Fig. 9, the utterances cover a variety of levels, from simple single-object identification to complex inter-object relations, including both spatial relationships and attribute recognition. In Fig. 10, the questions span from basic existential queries to more complex attribute-based and advanced inquiries. Finally, we show an example of 3D captioning for objects in the scene in Fig. 11. We can see that our model can produce reasonable results for these various cases.

**Failure Cases.** We subsequently show several failure cases for analysis. In Fig. 12, the model encounters challenges with understanding precise comparative spatial relationships (left panel) and occasionally predicts bounding boxes with imprecise sizes (right panel). Fig. 13 illustrates the model's shortcomings, which include generating incorrect information (hallucination), misunderstanding questions regarding object functions, and inaccuracies in object enumeration. Fig. 14 demonstrates the limitations of the LEO embodiment token design, which overlooks object dimensions and consequently struggles to identify overlapping elements, exemplified by the failure to distinguish between a shelf and a book resting upon it.

## D.5 In-the-Wild Test

To test the generalization capability of trained 3D grounding and language models, similar to EmbodiedScan, we use Azure Kinect DK to record the RGB-D streams with camera poses and feed them into our models. The question-answering test uses the improved PointLLM tuned with MMScan without any modification. The grounding test uses the trained EmbodiedScan baseline with the best performance, and we only visualize the top-k predictions matching the language descriptions, where k is adaptive according to the prompt, *e.g.*, if the question corresponds to a single target, we will

Table 13: Using VG and QA for tuning 3D-LLMs on traditional 3D question answering benchmarks, while "B-4" stands for "BLEU4", "M" for "METEOR", "R" for "ROUGE", and in "EM@1" we report the top-1 exact match and the refined exact-match protocol results respectively.

| Methods | ScanQA (val) | | | | SQA3D (test) |
|---|---|---|---|---|---|
| | B-4. | MET. | R.-L. | EM@1 | EM@1 |
| baseline | 10.5 | 15.1 | 39.2 | 23.1 (39.0) | 51.6 (54.1) |
| + captions | 12.7 | 19.8 | 48.1 | 24.7 (48.9) | 54.1 (56.8) |
| + VG samples | 12.0 | 16.3 | 42.1 | 24.1 (39.2) | 52.1 (55.2) |
| + QA samples | 16.2 | 17.6 | 49.4 | **27.9 (49.2)** | **54.9 (58.9)** |

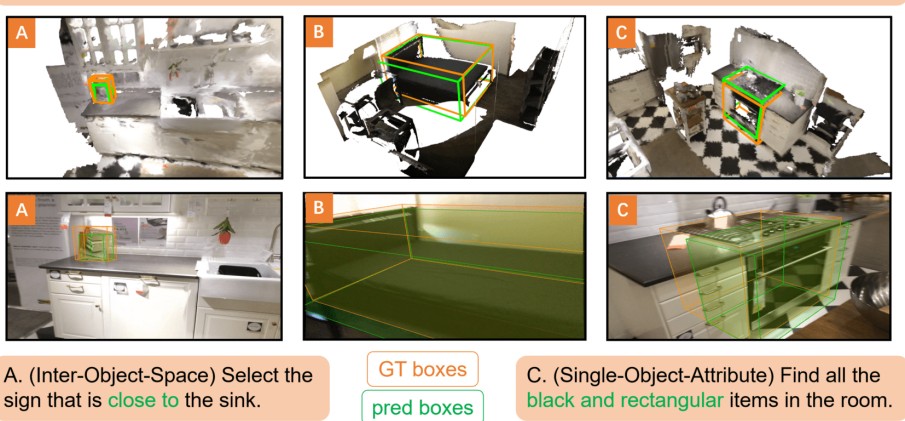

Figure 9: Visual grounding qualitative results, covering single and inter-target, spatial and attribute understanding.

only visualize the top-1 prediction. [2] It shows decent performance both in QA and VG regarding different aspects of language prompts, even with a different RGB-D sensor in unseen environments. We visualize the results in the attached supplementary video.

# E   Evaluation Details

This section presents more details regarding the GPT evaluation adopted in the main paper and further conducts human evaluation to validate the consistency of these two approaches.

## E.1   GPT Evaluation

**3D Captioning.** GPT4 is assigned to compare captions provided by humans with those generated by the model. During the labeling process, the captions are structured clearly to facilitate analysis. We identify common aspects such as Object Type, Color, Shape, Position, Function, and Design, and instruct GPT4 to assess whether these aspects are correct (scored as 1) or incorrect (scored as -1). A score of 0 indicates that the aspect is absent in the human captions and is thus omitted from evaluation. The evaluation prompt used is illustrated in Fig. 15.

**3D Question Answering.** Chat-GPT4 is required to compare answers provided by human and model-generated responses, identifying all key points in the human answers to serve as a reference for scoring. The evaluation results are then categorized into three classes: Correct, Ambiguous, and Error. 'Correct' indicates that the model's key points align with those of the human responses. 'Ambiguous' means it is unclear whether the key points are equivalent or incorrect. 'Error' denotes that the key points are either incorrect or missing. Additionally, a chain-of-thought technique is employed to prompt Chat-GPT4 to explain its responses. The prompt used is illustrated in Fig. 16.

---

[2]Given the practical use, it is important to explore a certain score threshold to meet general grounding requirements in the future.

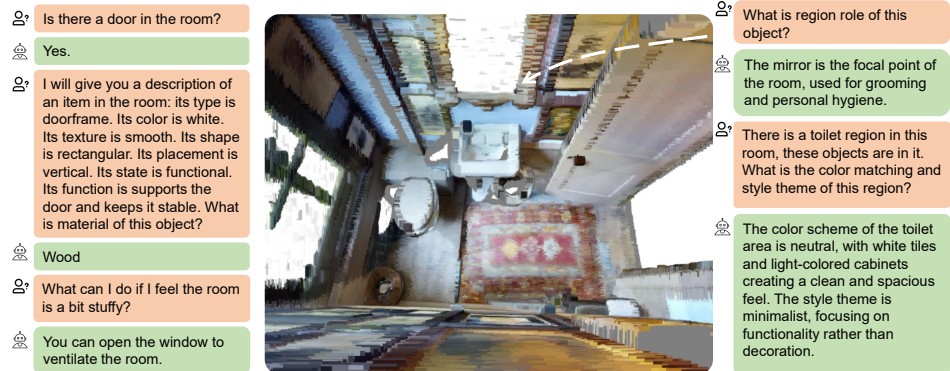

Figure 10: Question answering qualitative results, covering existential, attribute understanding, and advanced queries.

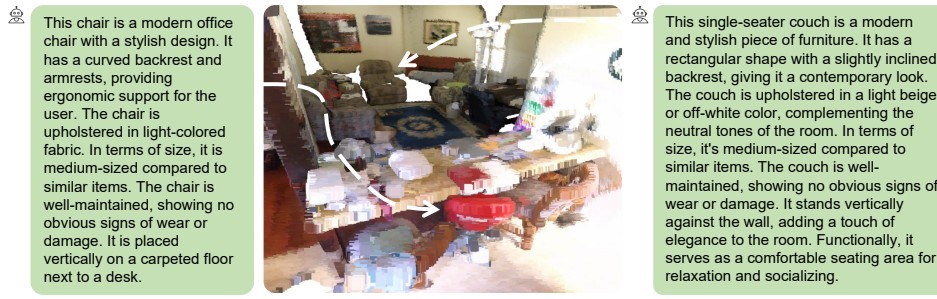

Figure 11: Qualitative results for 3D captioning.

## E.2 Human Evaluation

As mentioned in the main paper, we conduct human evaluation on the language tasks, including question-answering and captioning benchmarks, to validate the consistency of human and GPT evaluation.

**Question answering evaluation.** We randomly select 300 QA pairs from zero-shot 3D-LLM, Chat-3D-V2, LL3DA, and LEO models, including both fine-tuned LL3DA and LEO versions. We then recruit five well-educated human evaluators to assess each result based on specific guidelines provided.

The human evaluation guidelines for question answering are as follows:

*Please compare the ground truth answers and model-generated answer using the following metrics:*

*Hallucination: 0: Clear hallucination 1: No hallucination*

*Completeness: 0: Completely incorrect 1: Partial coverage 2: Complete coverage*

Table 14: Human evaluation results for the 3D question answering benchmark.

|  | Model | Hallucination | Completeness | Overall |
|---|---|---|---|---|
| Zero-shot | 3D-LLM | 33.1 | 25.7 | 29.4 |
|  | Chat3d-v2 | 26.7 | 33.1 | 29.7 |
|  | LL3DA | 27.2 | 21.3 | 24.3 |
|  | LEO | 32.7 | 26.3 | 29.5 |
| Fine-tuned | LL3DA | 67.0 | 63.7 | 65.3 |
|  | LEO | 71.7 | 70.2 | 70.9 |

We evaluate the fine-tuned LL3DA and LEO, with results presented in Tab. 14. These results show a similar trend with GPT evaluation shown in the main paper, validating the reliability of GPT evaluation.

**Caption evaluation.** Similarly, for the captioning task, we randomly select 300 captions from the fine-tuned LL3DA and LEO models. Subsequently, we hire five evaluators to assess each result,

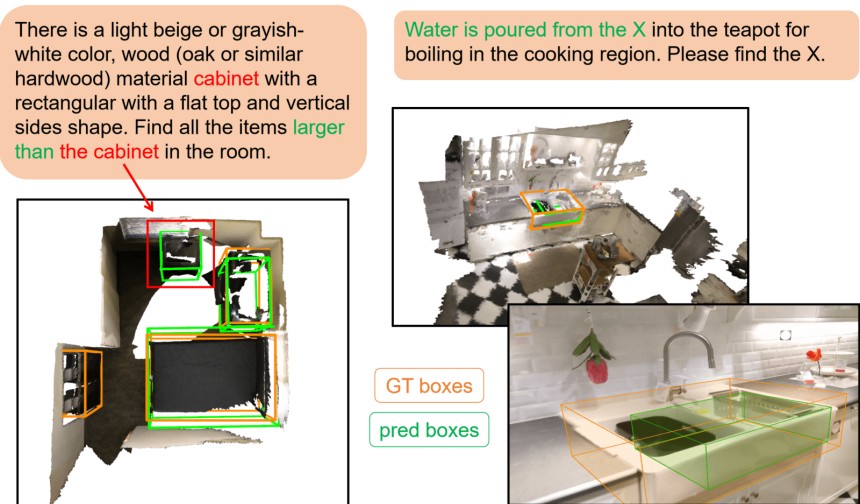

Figure 12: Vision grounding failure cases. **Left**: The model falsely selected the cabinet itself as an item larger than the specified cabinet, also missing the shelf on the left. **Right**: The model successfully understands the function of a sink, but predicts a box with inaccurate size.

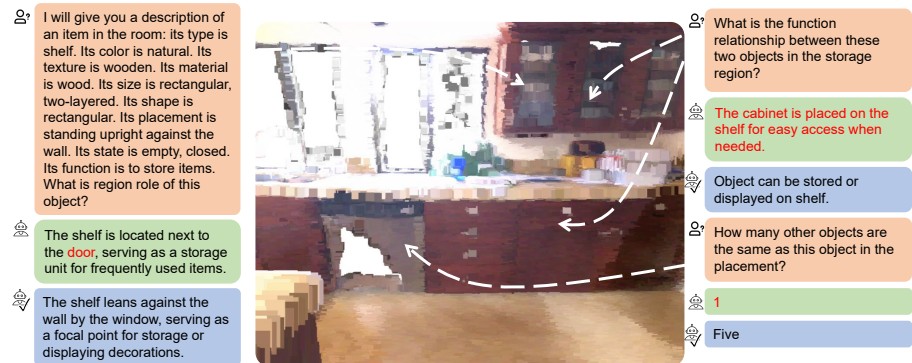

Figure 13: Question answering failure cases. Errors are marked in red.

using the same criteria provided to Chat-GPT4 as shown in Fig. 15. The results have been shown in Tab. 9 with the same consistent trend.

# F  Dataset Details

This section presents more dataset details, including more statistics on meta-annotations and the license and access issues.

## F.1  More Statistics

In the main paper, we only show the overall statistics for the dataset, such as the numbers of meta-annotations, VG and QA data samples, and grounded scene captions. We also include the composition information of post-processed data samples in the main paper's Fig. 3. Here, we further supplement the detailed word cloud comparison with EmbodiedScan and distribution statistics for objects and regions of meta-annotations in Fig. 17. We can observe that MMScan shows significantly better diversity regarding language annotations than EmbodiedScan and also covers most of the common object and region categories.

## F.2  License and Access

Our dataset is built upon EmbodiedScan [47], which collects the raw data from ScanNet, 3RScan, and Matterport3D. To access and use these three raw datasets, users should follow their original

**Human**

This is a wooden bookshelf, the focal object of the room. The bookshelf is of moderate size, rectangular in shape, and has a dark brown color. It is placed vertically against the wall and is well-maintained. It is used to store items. The design of the bookshelf is simple and practical, with no additional decorative features.

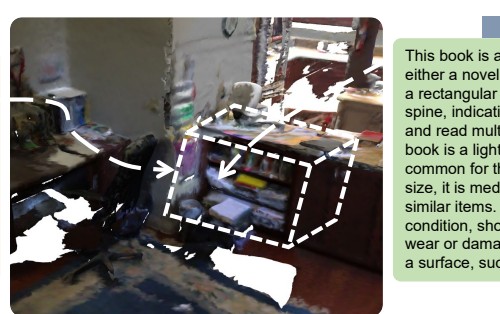

**Model**

This book is a standard-sized hardcover, either a novel or a reference book. It has a rectangular shape with a slightly tilted spine, indicating that it has been opened and read multiple times. The color of the book is a light beige or off-white, which is common for this type of book. In terms of size, it is medium-sized compared to similar items. The book is in good condition, showing no obvious signs of wear or damage. It is standing upright on a surface, such as a bookshelf or table.

Figure 14: Object caption failure cases. The model fails to understand the size of the bounding box.

Evaluate a model-generated object caption result against a human-generated caption for a 3D language model. Focus on the following aspects: object type, color, shape, position, function, and design. If the human caption does not include any of these aspects, mark it as 0; otherwise, use 1 or -1 to indicate true or false(missing or incorrect). Following the examples provided, complete the last row of the table:

| Human caption | Model caption | Object type | Color | Shape | Position | Function | Design |
|---|---|---|---|---|---|---|---|
| In the center of the room stands a spacious arched doorway, crafted from dark wood, exuding a strong sense of elegance. It is in excellent condition. Compared to the other items in the room, this door is quite large, vertically placed against the beige wall. The door is functional, providing access to another room. Its design features include dark wood veneer and an arched top, adding a touch of refinement to the room. | The doorframe is a prominent feature in the room. It is rectangular in shape and painted in a light beige or off-white color. Made of wood, it exudes a warm and inviting feeling. The doorframe is relatively large, taking up a significant portion of the wall space. In terms of functionality, it serves as a typical doorframe, connecting one room to another. Compared to similar items, this doorframe is well-maintained with no obvious signs of damage or wear. Its design is simple yet elegant, blending seamlessly into the overall aesthetic of the room. | 1 | -1 | 0 | -1 | 1 | -1 |
| This is a single object, a chair. The chair is white and made of plastic. It is positioned slightly tilted to the right on a light-colored floor. The chair is empty and well-maintained. Its design is simple, without armrests or a backrest. Compared to similar items, this chair is of medium size. Its function is to provide a comfortable seat, and it effectively fulfills this purpose. | This is a white, rectangular box. The box is made of material such as plastic or stainless steel, and it's in good condition with no obvious signs of wear or damage. It's placed vertically on the floor, and there are no other items next to it. Compared to similar items, this box is of medium size. Its design is simple and practical, without any extra features or decorations. | -1 | 1 | 0 | -1 | -1 | -1 |
| [human caption] | [model caption] | | | | | | |

Rules:
1. First, carefully read the human caption to identify the six key attributes, then compare them with the model caption.
2. Format the 6 scores separated by "|" (beginning and ending with "|"). Respond with only the formatted scores.

Figure 15: **Prompt for GPT evaluation.** The table is provided in Markdown format.

licenses [5, 3, 1] and ask their official hosts for authorization. It is typically smooth if only using them for academic research. For the annotations of EmbodiedScan and MMScan, we follow a simple approach for authorization by collecting Google forms [2] temporarily, with a simple license attached to this form. Due to this work being a follow-up of EmbodiedScan regarding language annotations, the code to reproduce baselines and experiments will also be released at `https://github.com/OpenRobotLab/EmbodiedScan`.

Evaluate a model-generated QA result against a human-generated answer for a 3D model. Begin by identifying all key points in the human answer, then compare each of these attributes in the model-generated answer, providing reasons for each evaluation. Following the examples provided, complete the last row of the table:

| Question | Human Answer | Model Answer | Total Key Points | Correct | Ambiguous | Error | Reason |
|---|---|---|---|---|---|---|---|
| What is the purpose of the object | Hanging clothes | Providing storage space | 1 | 1 | 0 | 0 | A place for hanging clothes also provides storage space. |
| What is the usage of the object | Support the feet | Suitable for daily use | 1 | 0 | 1 | 0 | "Suitable for daily use" is ambiguous and does not specify support for the feet. |
| There is a toilet region in this room, these objects are in it. What is the location and function description of this region? | This is a bathroom that focuses on cleanliness and grooming activities. It includes facilities such as a sink, toilet, and a storage vanity. | This area appears to be a bathroom or toilet area, specifically designed for personal hygiene and sanitation. | 3 | 2 | 0 | 1 | The model identifies the area as a bathroom and notes its usage, but misses specific objects like the sink and vanity. |
| Are these two objects the same in shape? Why? | No, the chair is irregular and the lamp is cylindrical. | Yes, the chair and the chair are both irregular. | 3 | 1 | 0 | 2 | The model incorrectly identifies both objects as chairs and fails to recognize the cylindrical shape of the lamp. |
| Question | Human Answer | Model Answer | | | | | |

Rules:
1. Focus on comparing the two answers.
2. Only include the completed last row of the table in your response, excluding the header.

Figure 16: **Prompt for GPT evaluation.** The table is provided in Markdown format.

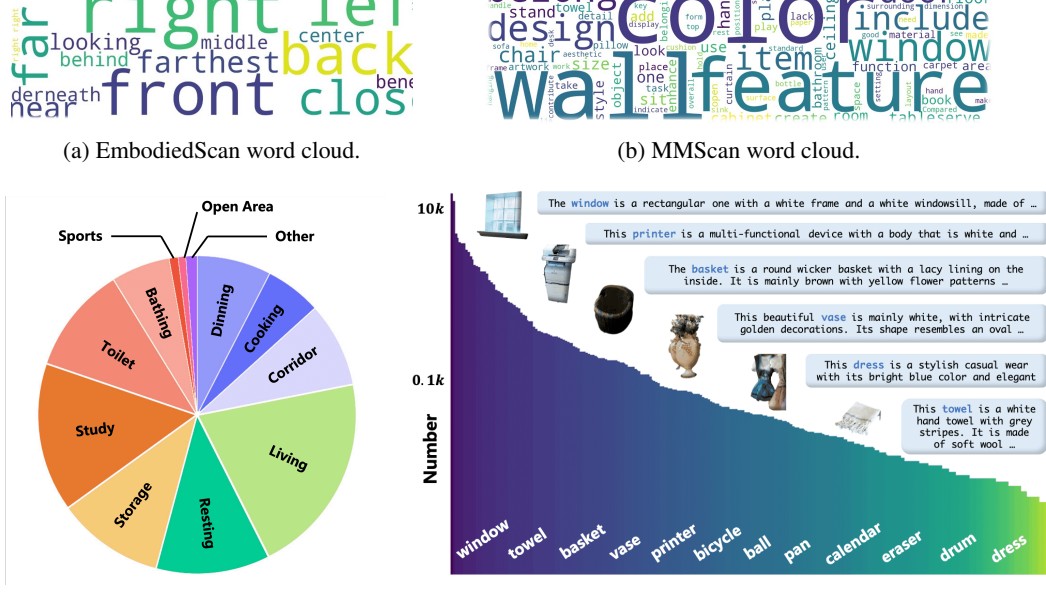

(a) EmbodiedScan word cloud.

(b) MMScan word cloud.

(c) Regions' category distribution.

(d) Objects' category distribution.

Figure 17: Statistics. (a)(b) Comparing the word clouds of EmbodiedScan and MMScan, we can observe the significant diversity improvement in the language annotations, from focusing on inter-object spatial relationships only to holistic understanding. (c)(d) The distributions of region and object annotations.

