# OpenReview forum: "MMScan: A Multi-Modal 3D Scene Dataset with Hierarchical Grounded Language Annotations"
_NeurIPS.cc/2024/Datasets_and_Benchmarks_Track — NeurIPS 2024 Track Datasets and Benchmarks Poster_

### Official Review · Reviewer_iBNL · 2024-07-22
**Review of MMScan**

**Rating:** 6
**Confidence:** 5
**Clarity:** Yes.

**Review:**

Pros:
* This work proposed a synthetic multi-modal 3D vision language (3DVL) dataset by annotating images captured in the indoor 3D scenes by using LLMs and VLMs.
* It demonstrated that training with this large-scale 3DVL dataset enhance the ability of 3DVL tasks such as 3D question answering and 3D visual grounding.


Cons:
* Most findings obtained from the dataset are already known from existing works 3D-VisTA, SceneVerse, and LEO.
* Many methods for creating 3D Language datasets using LLMs and VLMs have already been proposed (3D-VisTA, LEO, Multi3DRef, LEO). There is no novelty in dataset creation methods.
* Since human performance is not evaluated on this dataset, it is unclear whether the dataset is challenging or not.
* I could not ascertain the validity of the 3D language benchmark on MMScan, as the dataset has not undergone manual validation.
* The paper does not clearly state the license for the annotations of the MMScan dataset.

Other comments
* I reviewed the data at https://github.com/OpenRobotLab/EmbodiedScan but could not find the MMScan dataset and code. Despite reading the supplementary material, the reasons for not disclosing the dataset and code remain unclear.
* Table 1 shows that the ScanQA annotations are template-based, but this is incorrect as they have been edited by humans.
* Regarding Table 1, although RIORefer [1] and ARKitSceneRefer [2] have already been proposed as studies in 3D Visual Grounding, they are not cited as related work.

[1] Cross3DVG: Cross-Dataset 3D Visual Grounding on Different RGB-D Scans (3DV 2024)

[2] ARKitSceneRefer: Text-based Localization of Small Objects in Diverse Real-World 3D Indoor Scenes (Findings of EMNLP 2023)

So far, I'm leaning between weak accept and weak reject now.
Please let me know if I misunderstood the paper’s content.

**Strengths:**

The proposed MMScan dataset, containing 6.9M language annotations In the real-world 3D scenes is useful for training 3VLM models.

**Additional Feedback:**

None

**Correctness:**

The data collection method is appropriate, and the data creation process is well described, but the dataset lacks details on manual verification and human correction.

**Documentation:**

There is no mention of licensing and maintenance plans.

**Limitations:**

The authors mentioned the limitations and potential negative societal impact in their paper.

**Opportunities For Improvement:**

It is recommended to correct the mistakes identified in the Review section.

**Relation To Prior Work:**

Some relevant existing datasets are not mentioned.

**Summary And Contributions:**

The paper introduces a large-scale multi-modal 3D dataset, MMScan, aimed at advancing 3D vision language tasks using LLMs and VLMs. It also benchmarks these tasks using recent 3D visual grounding models on the MMScan dataset.

---

> ### Author Rebuttal · Authors · 2024-08-17
>
> Thank you for acknowledging our extensive 3DVL dataset and your questions about its distinctions from related studies, the validity of our benchmark and samples. Below are our detailed answers.
> ## W1: Different findings from existing works 3D-VisTA, SceneVerse, and LEO.
> We agree that there are common findings in Sec. 4.3, which is normal because all of the mentioned works, including ours, need to justify the value of datasets for training models on different tasks. Meanwhile, our analysis reveals many distinct insights, particularly in our new benchmarks:
>   - VG benchmark (Sec. 4.1):
>     - New challenges from diverse and complex prompts, 9-DoF box estimation, and varying numbers of grounding targets make lower performance.
>     - Single-target performance is generally lower, showing models excel in inter-target tasks, which previous works focused on.
>     - Significant AP-AR discrepancies in models like ViL3DRef indicate the difficulty of grounding target selection in the problem, while EmbodiedScan performs well in this aspect.
>   - QA benchmark (Sec. 4.2):
>     - Zero-shot performance is low, and the model's ranking is also different from previous benchmarks (3D-LLM performs a little better) due to our comprehensive evaluation both in data and metrics.
>     - Using our dataset as a complementary source for tuning these models enhances accuracy by over 25%.
>     - Single-target QA performance is better due to broader coverage in previous QA datasets, and can be further improved with our data training.
>     - Zero-shot Chat3D-v2 and fine-tuned LEO perform better in complex reasoning tasks (advanced QA).
>   - Findings in Analysis (Sec. 4.3):
>     - The basic thing to validate is the efficacy of training with MMScan and the scaling law on different downstream tasks, so we put this content in the main paper.
>     - We also offer supplementary experiments to study the scaling law regarding scene vs. data sample diversity, multi-view inputs & detection pretraining, and MMScan for second-stage instruction tuning in Sec. 4 in the supplementary materials. All these provide additional findings in other aspects.
> ## W2: Novelty in dataset creation methods.
>   Though all of these works, including ours, have utilized powerful VLMs for annotation, MMScan is unique and can bring new insights in the following aspects:
>   - Top-down annotation logic covering region/object level, single/inter-target descriptions, and spatial/attribute understanding.
>   - Systematic and customized annotation workflows for objects and regions:
>     - Carefully designed language/visual prompts to cover different aspects to obtain meta-annotations instead of direct benchmark samples.
>     - Detailed ablation for the performance of different VLM choices.
>   - Adaptable methods for deriving different benchmark samples from meta-annotations.
>   - Human-in-the-loop design ensures quality and minimal biases, with UI tools prompting explicit error identification, achieving a sub-5% error rate.
> ## W3: Human performance evaluation.
>   Thank you for your query. We randomly selected 20 scenes, with one question per sub-category in our benchmark, totaling 120 QA samples to test human performance, as shown in the table below. We use manual evaluation to ensure the results are accurate. **We can see that humans performed much better, and the benchmark is moderately challenging.**
> | Method | ST-attr | ST-space | OO-attr | OO-space | OR  | Advanced    | Overal      |
> |-|-|-|-|-|-|-|-|
> | LEO (finetune)    | 52.6        | 40.0        | 45.0        | 30.0        | 45.0        | 35.0        | 44.6        |
> | LL3DA (finetune)  | 45.0        | 40.0        | 31.6        | 40.0        | 50.0        | 45.0        | 42.8        |
> | Human     | 85.0        | 68.4        | 66.7        | 77.8        | 84.2        | 70.0        | 77.8        |
> ## W4: Manual validation for the benchmark validity.
> Thank you for your query. Please refer to the answer to W2 in the common questions.
> ## W5: License and dataset & code release.
> Thank you for your kind reminder. We have cleaned and checked our data in the past month and released several data samples in the mentioned GitHub repo for reference. We follow the license and the access authorization approach of EmbodiedScan (mentioned in Sec. 6.2 in the supplemental material), and will get all of these ready this month (release EmbodiedScan v2, including MMScan and newly added 5k scan annotations on ARKitScenes).
>
> For the code, since the benchmark for different 3D-LLMs uses their official code and it needs a lot of effort to reorganize them as a clean, structured open-source version to the community, we are still working on it to make it meet the quality of the current EmbodiedScan codebase and will release them ASAP after finishing the remaining experiments in the discussion phase. Specifically, we will release the strongest method, the improved version of PointLLM, to the community first as the baseline in one month and release the others gradually.
> ## Typo in Table 1 and add related studies like RIORefer and ARKitSceneRefer.
> Thank you for your advice. We revised Table 1 to include a more extensive comparative analysis **in the pdf attached to the common response.**
>
> RIORefer and ARKitSceneRefer, which are similar to ScanRefer but focus on 3RScan and ARKitScene, respectively, also have limitations in terms of quantity and scene diversity. We also add SceneVerse, which has more scenes, including 47K synthetic scenes, but overall, the richness and quantity of the corpus are relatively limited. In addition, their data is either purely human-generated or entirely machine-generated, while our approach is different. We employ a human-in-the-loop methodology, ensuring that every piece of data benefits from human insight and intervention. Finally, to supplement the scene diversity of MMScan, we are also incorporating an additional 5k ARKitScenes along with 50k to 100k synthetically generated scene datasets as continuing efforts.

---

> > ### Comment · Reviewer_iBNL · 2024-08-25
> >
> > Thank you for your reply. I'm glad some of your concerns have been addressed. It's unfortunate that the code wasn't provided during the review period. Regarding the license, I have already read Sec. 6.2 in the supplemental material. However, I'm still unclear about the license for this annotation data. Is it CC BY 4.0, as explicitly stated on this website?

---

> > > ### Author Response · Authors · 2024-08-25
> > >
> > > Thanks for your feedback. I'm glad that our response addresses some of your concerns.
> > >
> > > We have been working on sorting out the code these days and will try to provide a version (that may not be ready to release but may help you understand some details) in the next few days during the discussion period.
> > >
> > > For the license, sorry for the confusion caused. The license is CC BY 4.0 as you see on the website. We will make it clear in the revised version.

---

> > ### Author Response · Authors · 2024-08-30
> > **Code preview available**
> >
> > We have sorted out some of the code for data processing and several baselines in QA/VG benchmarks and temporarily uploaded them at https://anonymous.4open.science/r/MMScan-beta-preview/README.md. Please refer to the README files under each directory for more details.
> >
> > We will keep polishing the code and officially release it on GitHub when its code and structure are of better quality. In addition, we will also commit to maintaining and improving the benchmark with the community's feedback with continuous efforts.
> >
> > Hope this information helps.

---

### Official Review · Reviewer_qEsA · 2024-07-25
**Large-scale 3D dataset with language-annotation that is promising for the community**

**Rating:** 7
**Confidence:** 4

**Review:**

Pros:
- Clarity:
   - The paper was well-written
   - Terminologies and concepts were well-explained
   - Results are easy to understand
- Significance:
   - The proposed dataset is of interest of the research community.
   - Results shown on the benchmarks are promising
- Originality:
   - The dataset was built on a prior work. Contributions from prior works were made clear.

Cons:
- Clarity:
   - Some information can be included or explained better (see Clarity section)

**Strengths:**

- MMScan is the largest-scale 3D scene understanding dataset with language annotations. The contribution of this dataset can greatly benefit the training and evaluation processes of multi-modal 3D scene understanding models.
- MMScan features language annotations on both region levels and object levels, which is novel compared to prior works that only focus on object-level annotations.
- The paper evaluated different SOTA methods on the 2 benchmarks: visual grounding and 3DQA. MMScan results show the that this is a more challenging dataset compared to previous datasets for. these 2 benchmarks.
- Results when including the grounded scene captioning annotations for training/instruction tuning show the benefit of having captioning annotations as additional training signals.
- The proposed dataset is relevant to the interest of the 3D multimodal learning research community.

**Additional Feedback:**

Please consider presenting Table 6 in a way that can showcase better the benefit of MMScan as suggested above

**Clarity:**

The paper is well-written. All terminologies and concepts are clearly explained. Tables present are easy to understand. Table captions can be improved. For example, in table 4, adding that the non-shaded columns represent GPT-4 metric in the caption would make it more self-contained.

**Correctness:**

The dataset was meticulously designed, with both the use of LLMs and human annotators. Claims made in the paper are correct and backed with evidence.

**Documentation:**

The dataset is well-documented in the Supplemental material.

**Ethics:**

No ethical concern

**Limitations:**

The authors discussed limitations of the proposed dataset.

**Opportunities For Improvement:**

- It is quite confusing why the baseline method for Table 6 is PointLLM and not one of the methods shown in Table 4.
   - What is the reason for choosing PointLLM to demonstrate the benefit of having the captioning model?
   - Does the benefit hold for other baselines present in Table 4?
   - Why isn't PointLLM included in Table 4 since it is a relevant baseline for QA task?
- L329-340: Was MMScan used for instruction tuning only? What was PointLLM pre-trained on? It's unclear whether PointLLM was pre-trained on the object and region relationship data as baseline or PointLLM was trained on some other data, then using MMScan (object + region relationship data and captioning data for fine-tuning).
- An improvement that can potentially be made is:
   - A table with all the baselines from Table 4, first showing the baselines' best performance on ScanQA and SQA3D from prior works, then showing how pre-training on MMScan data generalizes zero-shot to ScanQA and SQA3D, then showing any kind of additional instruction tuning with MMScan helps with the performance on ScanQA and SQA3D. Ideally training on MMScan data would improve the performance of these baselines on ScanQA and SQA3D over their prior SOTA performance on these benchmarks. Table 6 attempts to show this, but is not quite there.

**Relation To Prior Work:**

The paper discussed prior works and set themselves apart from prior contributions clearly.

**Summary And Contributions:**

- The paper introduced a large-scale language-annotated 3D scene dataset, MMScan. Language annotations are organized in a hierarchical manner, segmenting the scene into regions and objects. These annotations are collected by both LLMs and human annotators.
- MMScan includes language annotations for individual objects and regions, object-object relationships and object-region relationships.
- This dataset supports 2 benchmarks: visual grounding and 3DQA. The paper evaluated different SOTA methods on both benchmarks. Results obtained demonstrate the benefit of MMScan.
- This dataset also offers grounded scene captioning annotations that facilitate both training and instruction tuning purposes.

---

> ### Author Rebuttal · Authors · 2024-08-17
>
> Thank you for recognizing that our work can greatly benefit multi-modal 3D scene understanding models, for its large scale, its novel hierarchical language annotations, and its challenging nature on benchmarks. Thanks as well for the valuable questions on table caption clarity and details of captioning baseline methods. Below are our replies to these questions.
>
> ## W1: Table caption clarity can be improved.
> Thank you for pointing out the clarity issue. We will polish the table captions as suggested.
>
> ## Q1: More details about the baseline method in line 329-340?
> - PointLLM is one of the pioneering 3D-LLMs incorporating point clouds into LLMs, which targets **object-level** understanding due to the limited data at that time. To make it better fit the larger-scale scene-level understanding, we mainly improved the **encoder** in PointLLM, with **other training details remaining the same**. Our improved version takes the multi-view images as input and extracts the 2D patch-wise features per image via CLIP, and aggregates these patch-wise features by projecting them to 3D space via depth maps and camera parameters. The features go through the projection layer and are sent to LLM.
> - Similar to PointLLM and LLaVA, our baseline model **follows the same two-stage training**. In the first stage, we use caption data based on ScanNet from SceneVerse to train the projection layer, and then we conduct the instruction tuning on ScanQA and SQA3D in the second stage.
> - In the experiments for Table 6 of the main paper and Table 4 of the supplementary material, we respectively **add the caption data** (meta-ann. captions, scene captions, all captions) from MMScan in the **first stage** and the **VG/QA samples** in the **second stage** to validate the efficacy of MMScan's data in different stages of tuning 3D-LLMs. We will clarify these details in the final version.
>
> ## Q2: The baseline methods in Table 4 and 6 do not intersect?
> Our improved version of PointLLM **performs the best in our preliminary experiments**, so we take it as the baseline in the ablation study in Table 6. Here, we supplement its benchmark results (that should be presented in Table 4) for reference as follows. Due to the resources and training time limits, we train our baseline model on 1/10, 1/4, and 1/2 MMScan training datasets, respectively, the results show that even when trained on 1/10 training datasets, the performance of our baseline model achieves comparable, even better results compared with LL3DA and LEO, which are trained on the full MMScan.  Besides, with the increment of the training dataset ratio, our baseline model achieves consistent performance improvement and achieves SOTA.
>
> | Settings | GPT-Overall | SimCSE | S-BERT | B-1. | B-4. | R.-L | MET. | EM@1|
> |-----------|---------------|-----------|-----------|------|------|-------|-------|--------|
> | 1/10 Fine-tuning  | 48.1         | 70.6    |  71.7  |  36.8 |  11.2 |  51.7 | 17.9 | 40.2 (46.2)|
> | 1/4 Fine-tuneing  | 53.0         | 73.7    |  76.3  |  38.7 |  12.2 |  55.5 | 19.5 | 44.2 (51.4)|
> | 1/2 Fine-tuning   | 54.6         | 74.1    |  76.9  |  39.2 |  12.7 |  56.1 | 19.9 | 45.4 (52.1)|
>
> ## Q3: Suggestions for improving Table 6, more results of training with MMScan?
> Thank you for your valuable advice. We reimplement LEO and LL3DA using only ScanQA, SQA3D, and our caption data following their officially released code, as shown in the table below (also using the same EM@1 metric), and observe consistent improvement brought by tuning with MMScan.
>
> Table 1: ScanQA results.
>
> | Method            | baseline     | tune w/ MMScan |
> |-------------------|--------------|----------------|
> | Improved PointLLM | 23.1         | 24.7           |
> | LEO               | 16.94        | 17.54          |
> | LL3DA             | 15.61        | 16.83          |
>
> Table 2: SQA3D results.
>
> | Method                    | baseline     | tune w/ MMScan |
> |---------------------------|--------------|----------------|
> | Improved PointLLM         | 51.6         | 52.7           |
> | LEO                       | 48.50        | 51.13          |
> | LL3DA                     | 45.11        | 45.35          |

---

> > ### Comment · Reviewer_qEsA · 2024-08-31
> >
> > I appreciate the authors for their effort in addressing my comments. I will keep my initial score.

---

> > > ### Author Response · Authors · 2024-08-31
> > >
> > > Thanks for your feedback. We will take your suggestions and improve them in the final version.

---

### Official Review · Reviewer_GsMX · 2024-07-29
**A Multi-Modal 3D Scene Dataset for Visual Grounding**

**Rating:** 9
**Confidence:** 5
**Clarity:** Yes, the paper is well written and cl…

**Review:**

Pros
- (1) Creating the 3D scene dataset that supports 3D visual grounding models.
- (2) Satisfied the need for large scale, high quality 3D language datasets to advance multi-modal 3D perception.
- (3) The combination of VLM based language annotation and human in the loop refinement.
- (4) Coverage for both spatial and attribute understanding and annotations.
- (5) Open-source dataset to benefit the research community.

Cons
- (1) Lack of benchmark and comparison to recent related 3D language datasets

**Strengths:**

The main strength is that the introduction of MMScan satisfies the need of comprehensive and high quality 3D-language datasets.  This new benchmark dataset brings tremendous opportunities for training 3D perception models.

**Additional Feedback:**

NA

**Correctness:**

Yes, the claims and methods made are solid and the experiment details support the claims for the dataset.

**Documentation:**

There are sufficient details on dataset collection, availability and maintenance. The code and dataset have already been released. I checked the github repo, and found the code quality is good with good documentation.

**Ethics:**

No ethical concerns.

**Limitations:**

The authors shared some limitations, including relying on human annotators. The paper did not talk much about the societal impact.

**Opportunities For Improvement:**

The paper lacks discussion of potential dataset bias. In addition, it lacks comparison with latest related work (e.g. Sceneverse [1]). It also misses the details on annotation efficiency and comparison between fully manual annotation vs human in the loop annotation.

[1] Jia, Baoxiong, et al. "Sceneverse: Scaling 3d vision-language learning for grounded scene understanding." arXiv preprint arXiv:2401.09340 (2024).

**Relation To Prior Work:**

The authors claimed the novelty of proposed top down annotation method, but it does not have detailed comparison with some very recent works.

**Summary And Contributions:**

The main contributions include (1) introducing a top-down annotation framework to create holistic and hierarchical annotations for 3D scenes; (2) large scale 6.9M language annotations across 5K scenes, 109k objects and 7.7k regions; (3) creating a new benchmark for 3D visual grounding and question-answering tasks.

---

> ### Author Rebuttal · Authors · 2024-08-17
>
> Thank you for recognizing that our work meets the demand for comprehensive and high-quality 3D-language datasets, and brings tremendous opportunities for training 3D perception models. We will further improve our paper following the precious comments with details elaborated as follows.
>
> ## Q1: Discussion of potential dataset bias.
> Thank you for the concern about dataset bias. Please refer to the answer to W1 in the common response.
>
> ## Q2 (W1): Comparison with the latest related datasets.
> Thank you for your kind reminder. We have revised Table 1 to include a more extensive comparative analysis, with additional synthetic datasets, as shown below. The initial version did not include these works because we previously concentrated on the comparison with other real-scanned datasets.
>
> RIORefer and ARKitSceneRefer, which are similar to ScanRefer but focus on 3RScan and ARKitScene, respectively, also have limitations in terms of quantity and scene diversity. SceneVerse has many more scenes, as it contains 47K synthetic scenes, but overall, the richness, quantity, and granularity correspondence of the corpus are relatively limited. In addition, their data is either purely human-generated or entirely machine-generated, while our approach is different. We employ a human-in-the-loop methodology, ensuring that every piece of data benefits from human insight and intervention. Finally, to supplement the scene diversity of MMScan, we are also incorporating an additional 5k ARKitScenes along with 50k to 100k synthetically generated scene datasets as continuing efforts.
>
> Table 1: Comparison with other datasets.
> | Dataset            | #Scans | #Language | #Tokens | Real Scan | Correspondence   | Focus              | Annotation      |
> |--------------------|--------|-----------|---------|-----------|------------------|--------------------|-----------------|
> | ScanRefer          | 0.8k   | 52k       | 1.18M   | √         | Sent.-Obj.       | Natural            | Human           |
> | NR3D               | 0.7k   | 42k       | 0.62M   | √         | Sent.-Obj.       | Natural            | Human           |
> | SR3D               | 0.7k   | 115k      | 1.48M   | √         | Sent.-Obj.       | OO-Space           | Template        |
> | ScanQA             | 0.8k   | 41k       | -       | √         | Sent.-Obj.       | QA                 | AutoGen+Human   |
> | SQA3D              | 0.7k   | 33.4K     | -       | √         | Sent.-Obj.       | Situated QA        | Human           |
> | ScanScribe         | 3k     | 278K      | 18.49M  | √         | Sent.-Obj.       | Description        | GPT             |
> | Multi3DRef         | 0.7k   | 62K       | 1.2M    | √         | Sent.-MultiObj.  | Multi-Obj.         | GPT+Human       |
> | EmbodiedScan       | 5.2k   | 970k      | -       | √         | Sent.-Obj.       | OO-Space           | Template        |
> | RIORefer           | 1.4k   | 63.6k     | 0.94M   | √         | Sent.-Obj.       | Natural            | Human           |
> | ARKitSceneRefer    | 1.6k   | 15.6k     | 0.22M   | √         | Sent.-Obj.       | Natural            | Human           |
> | Sceneverse         | 68k    | 2.5M      | -       | Mixed     | Sent.-Scene./Obj. | Caption + OO-Space | GPT/Temp./Human |
> | MMScan (Ours)      | 5.2k   | 6.9M      | 114M    | √         | Phrase-Obj./Reg. | Holistic           | GPT+Temp.+Human |
>
> ## Q3: Details on annotation efficiency and comparison between fully manual annotation vs human-in-the-loop annotation.
> Thank you for your valuable suggestion. We have conducted a detailed analysis comparing the annotation efficiency as well as the quality between fully manual annotation and human-in-the-loop annotation. We selected 20 objects and 2 regions from some representative scenes for annotation for this purpose and compared the annotation efficiency quantitatively (with similar quality) in the table below. The comparison results show an overall increase in effective annotations with significantly less time spent, which proved that "human-in-the-loop annotation" can help improve annotation efficiency.
>
> Note: We also attach the statistics of all the scenes with human-in-the-loop as a reference for average tokens/entries. The average number of tokens indicates the length of a sentence, and the average number of entries indicates the number of sentences annotated for a specific region.
>
> Table 2: Comparison on the object-level annotation.
> | Metric            | Time (s/object)     | Avg Tokens (tokens/object) |
> |-------------------|--------------|----------------|
> | Sample Scenes (fully manual) | 64.5 | 44.6 |
> | Sample Scenes (human-in-the-loop) | 36.6 | 85.5 |
> | All Scenes (human-in-the-loop)         | -      | 91.8  |
>
> Table 3: Comparison on the region-level annotation.
> | Metric            | Time (s/region)     | Avg Entries (entries/region) | Avg Tokens (tokens/entry) |
> |-------------------|--------------|----------------|--------------|
> | Sample Scenes (fully manual) | 1155.9 | 16.0 | 19.8 |
> | Sample Scenes (human-in-the-loop) | 739.0 | 22.0 | 21.8 |
> | All Scenes (human-in-the-loop)         | -      | 23.9 | 25.0 |

---

> ### Comment · Reviewer_GsMX · 2024-08-31
> **Thanks for the very informative response, I maintain strong accept rating**
>
> I appreciate the authors taking time to respond to all concerns, including (1) dataset bias, (2) comparison with the latest work, and (3) efficiency comparison between human-in-the-loop annotation and manual annotation. I believe the responses from the authors are sound and add significant weight to the work.  It is well-written and informative response to the concern about dataset bias. For VLM-related biases and human annotator biases, they identified and addressed it from multiple perspectives: a. perception bias. b. human understanding bias. c. statement bias.
>
> For the authors, I encourage incorporating (1) the writing about dataset bias into the supplementary material, and putting (2) and (3) into the main body of the paper. Particularly, (3) shows a 36.6s vs 64.5s efficiency gain for per object, and 739s vs 1155.9s efficiency gain for per region. This should be highlighted.

---

> > ### Author Response · Authors · 2024-08-31
> > **Thanks for the feedback. We will update these content in the final version.**
> >
> > Thanks for your insightful suggestions in the review and the prompt feedback for our response. We agree that these results are important for strengthening this paper.
> >
> > We will add these contents to the final version of the main paper and supplemental materials as suggested.

---

### Official Review · Reviewer_bd7y · 2024-08-01

**Rating:** 5
**Confidence:** 4
**Correctness:** Yes.
**Clarity:** Yes.

**Review:**

The work presents a contribution to the field of 3D scene understanding.  The paper seems to be well-structured, providing detailed information. However, there are some areas where more clarity could be beneficial, particularly regarding the annotation process.

Pros:
1. Benchmark establishment: Provides two major benchmarks for 3D visual grounding and question-answering.
2. Experimental validation: Demonstrates the dataset's utility through model training and performance improvements. However, it may be insufficient.
3. Addresses existing limitations: Aims to overcome shortcomings in current 3D scene datasets.

Cons:
1. Unclear annotation process: Lacks detailed information on manual annotation and correction procedures.
2. Incomplete method comparison: Does not test against a wide range of existing methods in the 3D scene understanding field.
3. Absence of joint image modality: Lacks integration with image data, which is common in existing methods.
4. Potential biases: More information is needed on how annotation biases were identified and mitigated. Moreover, the usage of LLM can introduce biases.

**Strengths:**

- Comprehensive Dataset: MMScan offers a larger and more comprehensive dataset compared to existing multimodal 3D scene datasets. It includes 1.4 million meta-annotated descriptions across 109k objects and 7.7k regions.
- Benchmark Establishment: The paper not only introduces the dataset but also establishes benchmarks for 3D visual grounding and question-answering, providing valuable resources for evaluating and advancing 3D-LLMs.
- Experimental Validation: The authors validate the dataset's effectiveness by training models and demonstrating performance improvements in existing benchmarks and real-world applications.

**Additional Feedback:**

N/A.

**Documentation:**

I can not find the code in the main paper.

**Ethics:**

N/A.

**Limitations:**

Yes.

**Opportunities For Improvement:**

1. Unclear Annotation Process: The authors do not provide detailed information on how manual annotation and correction were conducted, including the extent of human intervention. This information is crucial for assessing the dataset's value.

2. Limited Scene Diversity: The dataset's scene diversity is limited, which is a significant ongoing issue in the field that remains unresolved. This may restrict the generalization ability of models trained on MMScan.

3. Testing of Relevant Methods: 3DVG is a mature field, yet the authors did not test a wider range of benchmark methods.

4. Can the authors provide more details about the types of biases observed during the human annotation process and how these were mitigated? Also, how to solve the biases introduced by LLM?

5. What specific steps were taken to ensure the comprehensiveness and accuracy of annotations, especially in complex scene contexts?

6. The dataset lacks joint image modality, which is commonly used in existing methods.

**Relation To Prior Work:**

Yes.

**Summary And Contributions:**

This paper introduces MMScan, a multimodal 3D scene dataset featuring hierarchical grounded language annotations. It is highlighted as the largest dataset in this category, containing 6.9 million annotations covering various aspects of 3D scenes, including object-level and region-level details. The dataset aims to address the limitations of existing 3D scene datasets by providing comprehensive annotations for visual grounding and question-answering benchmarks. The paper also details the dataset creation process, including data collection, annotation, post-processing, and the establishment of evaluation benchmarks. Additionally, it presents experimental results demonstrating the dataset's utility in training 3D visual grounding and question-answering models, and discusses scaling laws in multimodal 3D learning.

---

> ### Author Rebuttal · Authors · 2024-08-17
>
> Thanks for recognizing our contribution to the field of 3D scene understanding. The main questions center around the annotation process and validity of MMScan. We address them in detail as follows.
> ## W1 (Q1 & Q5): Unclear Annotation Process. What specific steps were taken to ensure the comprehensiveness and accuracy of annotations, especially in complex scene contexts?
> Thank you for your query. Please refer to the explanation for W2 in the common response.
>
> ## Q2: Scene Diversity.
>
> I agree with your comments regarding the importance of the scenes' diversity, and we are also making continuing efforts to address this problem. Our MMScan dataset is currently based on the 5k scans from the EmbodiedScan, which is one of the largest collections of existing real-scanned open datasets. We originally focused on these real-scanned data to reduce the potential sim2real gap when deploying the model trained with our dataset. In addition, we have also completed the EmbodiedScan's annotation on ARKitScenes and will include their MMScan's annotation in the following two months, leading to about **doubling** the number of real-scanned scenes. Meanwhile, we have **another 50k-100k photo-realistic synthetic scenes** to be annotated following the same pipeline to supplement the interactive scene data. We believe such continuing efforts will also amplify the value of MMScan's annotation pipeline.
>
> ## W2 (Q3): Testing of Relevant VG Methods.
>
> Thank you for your query. Due to the limited resources, previously, we only conducted baselines that had divergent approaches for the benchmark in the main paper. We are supplementing more methods, including MVT, 3D-VisTA, and ReGround3D, as shown in the table below, and observe a similar discrepancy between AP and AR results as those in our paper, indicating the challenge of grounding target selection from detecting objects. More methods will be benchmarked in the final release.
> | Methods      | AP25  | AR25  | AP50 | AR50  | ST-attr | ST-space | OO-attr | OO-space | OR   |
> |:------------:|:-----:|:-----:|:----:|:-----:|:-------:|:--------:|:-------:|:--------:|:----:|
> | ReGround3D [1]  | 4.12  | 48.12 | 1.98 | 22.12 | 4.23    | 3.98     | 7.32    | 6.98     | 8.23 |
> | MVT [2]         | 3.65  | 72.38 | 1.02 | 51.50 | 1.74    | 2.34     | 3.58    | 4.45     | 1.49 |
> | 3D-VisTA [3]    | 5.24  | 72.51 | 1.91 | 51.85 | 4.91    | 4.39     | 5.75    | 5.99     | 6.35 |
> | EmbodiedScan [4] | 10.49 | 47.21 | 2.94 | 21.76 | 7.44    | 7.53     | 13.65   | 11.19    | 7.74 |
>
> [1] ScanReason: Empowering 3D Visual Grounding with Reasoning Capabilities, ECCV 2024
>
> [2] Multi-View Transformer for 3D Visual Grounding, CVPR 2022
>
> [3] 3D-VisTA: Pre-trained Transformer for 3D Vision and Text Alignment, ICCV 2023
>
> [4] EmbodiedScan: A Holistic Multi-Modal 3D Perception Suite Towards Embodied AI, CVPR 2024
>
> ## W4 (Q4): Discussion of biases in the dataset.
> Thank you for your reminder. Please refer to the answer to W1 in the common response.
>
> ## W3 (Q6): Lack joint image modality.
> Thank you for your comment. Similar to EmbodiedScan, **our dataset incorporates a comprehensive integration with the image data**. In addition to the corresponding point clouds for each scene, there are also a certain number of multi-view RGB-D images covering the entire scene. These images are pivotal in **initializing the meta-annotations** for objects and regions. Furthermore, workers must modify the annotations with direct reference to these corresponding images, ensuring a robust and well-coordinated annotation workflow.
>
> For our **benchmarks**, our approach within the current setting is focused on inputs consisting of 3D point clouds with "colors," which can be interpreted as painted images onto point clouds. We will supplement the clarification of this detail in the final paper version. Finally, in Table 3 of the supplementary material, we also conducted preliminary experiments using multi-view image inputs. We will consider adding more experiments studying the 3D scene understanding problem based on the image modality.

---

### Author Rebuttal · Authors · 2024-08-17

Thank you for all your valuable suggestions. We have identified common questions and provided answers below.
# 1. Potential biases in the dataset.
 Bias in datasets generally stems from **uncertain** annotation guidelines or situations where **subjective judgment** is heavily relied upon. To address this, we designed a **top-down logic** to decompose and refine the annotation requirements, thereby enhancing overall comprehensiveness and reducing freeform annotations. This approach **systematically reduces bias** compared to previous works. However, some bias is inevitable during annotation. In our screening of VLM and manual annotations, we identified and addressed these biases as follows.
## Bias from VLMs:
- **Perception Bias**: Bias may occur due to **incomplete image inputs** (e.g., a cropped image might cause the VLM to focus on the object itself but overlook its relationship with the surrounding environment). To address this, we provide both cropped images and global images with the target object highlighted. At the region level, we supply several images taken from good angles that include all pertinent objects.
- **Understanding Bias**: VLMs might carry **stereotypes** (e.g., frequently describing a chair with "armrests" even if the chair does not have any) or misunderstand requirements (e.g., describing unrelated objects). Human annotators correct these by comparing images and revising errors.
- **Statement Bias**: VLMs often use **uncertain terms** like “possibly” or “seems,” which are inappropriate for definitive annotations. We screen these with templates and have annotators rewrite vague expressions.
## Bias from Human Annotators:
- **Perception Bias**: To mitigate the bias similar to that in VLMs, annotators receive multiple images, bird's eye view, and touchpoint display images, as shown in the attached pdf, to help them form a 3D concept of the scene and have enough observations from different perspectives to grasp detailed information.
- **Understanding Bias**: Annotators might be careless or make errors (e.g., they might overlook an aspect of an object's appearance, leading to incomplete descriptions, or they might fail to correct errors in VLM results).
- **Statement Bias**: Human annotations are expected to be natural and professional, but **ambiguities and incoherent expressions** can arise when rewriting VLM results (e.g., modifying the description of color in one place but not another, causing ambiguity, or omitting the subject of a sentence after edits). Additionally, since the workers may not be proficient in English, all our user interfaces are designed to support Chinese, helping the workers better understand and express the required annotations, but we must avoid bias during the translation process. **We address these issues through multiple rounds of sampling, checking, and feedback.**
# 2. Details on manual annotation and correction to ensure the validity of the dataset.
Our MMScan pipeline includes two stages: meta-annotation -> sample derivation for benchmarks. Below, according to the reviews, we particularly supplement more details of the manual annotation and correction process, which ensures the validity of our dataset and benchmark. We first present the details of the overall human-in-the-loop pipeline and then elaborate on the UI designs for this process.
## Manual Annotation and Correction Pipeline:
To ensure the quality of the final annotations, we adhere to the 95% principle. We randomly inspect 5%-10% of annotations in each screening round. If the pass rate is below 95%, **we identify issues, have annotators make corrections, and re-inspect** until the standard is met. Finally, we conducted one round of manual annotation and sampling verification for region segmentation. For object-level annotation and region-level annotation, we performed three rounds of manual annotation and sampling verification. Ultimately, each of these met the 95% quality requirement.

For benchmark samples, we derive them following several templates (for different aspects of questions shown in Fig. 3 in the main paper) and then enrich the questions' diversity by refining them with ChatGPT. The overall process introduces **very few biases** and just **relates to reorganizing the information**, avoiding obvious risks of yielding factual errors. We verify the logic of derived samples and ensure the overall quality **following the previously mentioned random inspection and 95% principle**. It turns out that all the samples meet our requirements well.
## UI design for region segmentation annotation:
At the region segmentation annotation stage, we expect workers to divide each scene into regions and label them with corresponding categories (living, cooking, resting, etc.). We designed a user interface (UI) to facilitate this process (**shown in the pdf**). The UI provides workers with a bird's eye view of the scene as well as the best shots of each object within the scene. When the worker touches a position in the BEV, the best shots of the object in the corresponding position will be automatically given. Workers can then delineate polygons by clicking on the BEV to segment the regions and assign them appropriate categories.
## UI design for the manual check of meta-annotations
We also have customized UIs for the manual check in the meta-annotations due to the complexity. At this stage, we expect workers to check and correct the VLM-generated meta-annotations for both format and content. We have designed two separate UIs for these modules (**shown in the pdf**), providing the workers with the best images of the corresponding objects/regions and the initial descriptions generated by the VLM. Workers need to verify the accuracy of the initial descriptions in multiple specified aspects for both objects and regions and rewrite and correct sentences generated by the VLM that do not conform to the required format, are factually incorrect, or are unnatural or problematic in expression.

---

### Decision · Program_Chairs · 2024-09-26

**Decision:**

Accept (Poster)

**Comment:**

This paper obtained positive recommendations from all reviewers, including one strong accept. All the reviewers appreciated the proposed systematic benchmarks for 3D visual grounding and question-answering. The main concerns were unclear novelty, missing details, and lack of baseline experiments, but the authors provided successful rebuttals in detail, assuaging the concerns. AC thus recommends acceptance of this submission and encourages the authors to incorporate the discussions in the final manuscript.